# Lung extracellular matrix modulates KRT5+ basal cell activity in pulmonary fibrosis

Richard J. Hewitt [1,2], Franz Puttur[1], David C. A. Gaboriau [3], Frédéric Fercoq [4], Maryline Fresquet[5], William J. Traves[1], Laura L. Yates[1], Simone A. Walker[1], Philip L. Molyneaux [1,2], Samuel V. Kemp [2,7], Andrew G. Nicholson[2], Alexandra Rice[2], Edward Roberts [4], Rachel Lennon [5], Leo M. Carlin [4,6], Adam J. Byrne[1], Toby M. Maher[1,8] & Clare M. Lloyd [1] ✉

Aberrant expansion of KRT5+ basal cells in the distal lung accompanies progressive alveolar epithelial cell loss and tissue remodelling during fibrogenesis in idiopathic pulmonary fibrosis (IPF). The mechanisms determining activity of KRT5+ cells in IPF have not been delineated. Here, we reveal a potential mechanism by which KRT5+ cells migrate within the fibrotic lung, navigating regional differences in collagen topography. In vitro, KRT5+ cell migratory characteristics and expression of remodelling genes are modulated by extracellular matrix (ECM) composition and organisation. Mass spectrometry-based proteomics revealed compositional differences in ECM components secreted by primary human lung fibroblasts (HLF) from IPF patients compared to controls. Over-expression of ECM glycoprotein, Secreted Protein Acidic and Cysteine Rich (SPARC) in the IPF HLF matrix restricts KRT5+ cell migration in vitro. Together, our findings demonstrate how changes to the ECM in IPF directly influence KRT5+ cell behaviour and function contributing to remodelling events in the fibrotic niche.

Single-cell RNA-sequencing has provided unique insight into the cellular complexity of the lung in idiopathic pulmonary fibrosis (IPF); a progressive and incurable scarring lung disease[1–3]. It has revealed novel cell phenotypes associated with dramatic changes to the lung microenvironment which are characteristic of this disease[1,2]. In IPF, normal alveoli are obliterated by a dense fibrous matrix causing architectural remodelling and the formation of distorted, cystic airspaces termed honeycomb cysts (HCs)[4,5]. In fibrotic lung, basal cells (BCs) expressing cytokeratin 5 (KRT5) – which normally reside in the conducting airways[6] - extend into the alveolar space and line HCs[5,7–9]. The presence of KRT5+ BCs in the alveolar compartment in IPF has been associated with increased mortality[10]. Studies have focused on delineating the origin of KRT5+ BCs in IPF[11,12] but little is known about how these cells interact with the lung extracellular matrix (ECM) to contribute to disease pathology.

ECM provides a physical scaffold for the adhesion of resident cells and hosts a complex array of biochemical and mechanical signals that influence cell phenotype and behaviour[13]. Composed of around 1000 structural and regulatory proteins, the human matrisome[14] is highly dynamic and undergoes remodelling[15]. Within the lung, ECM is produced and secreted by fibroblasts, of which there are multiple subpopulations with distinct anatomical localisation[16]. ECM

[1]National Heart and Lung Institute, Imperial College London, London SW7 2AZ, UK. [2]Royal Brompton and Harefield Hospitals, Guy's and St Thomas' NHS Foundation Trust, London SW3 6NP, UK. [3]Facility for Imaging by Light Microscopy, National Heart and Lung Institute, Imperial College London, London SW7 2AZ, UK. [4]Cancer Research UK Beatson Institute, Glasgow G61 1BD, UK. [5]Wellcome Centre for Cell-Matrix Research, Division of Cell-Matrix Biology and Regenerative Medicine, School of Biological Sciences, Faculty of Biology Medicine and Health, The University of Manchester, Manchester M13 9PT, UK. [6]School of Cancer Sciences, University of Glasgow, Glasgow G61 1QH, UK. [7]Present address: Department of Respiratory Medicine, Nottingham University Hospitals NHS Trust, City Campus, Hucknall Road, Nottingham NG5 1PB, UK. [8]Present address: Keck Medicine of USC, 1510 San Pablo Street, Los Angeles, CA 90033, USA. ✉e-mail: c.lloyd@imperial.ac.uk

dysregulation in disease states, compromises the activity of many different cells and processes, disturbing tissue homeostasis with a profound effect on organ function[15]. ECM alterations during IPF have been shown to directly influence fibroblast activity through mechanosignaling pathways, catalysing the pathobiology of the disease[17–19].

Epithelial cells acquire an enhanced migratory phenotype during tissue development, repair, cancer and fibrosis[20]. Cell migration is a key aspect of immune cell function within lung tissue and previous work from our laboratory has shown it is highly dependent on cues from the ECM[21]. ECM properties including molecular composition and topology likely influence cell motility[22]. Understanding mechanisms of epithelial cell redistribution during fibrotic lung remodelling is critical in the development of therapeutics that may help to preserve adequate gas exchange.

Herein we utilise human lung tissue, primary cells and cell-derived matrices to show how changes in the properties of the ECM during fibrotic remodelling dictate KRT5+ BC distribution and migration. We reveal that the local tissue microenvironment of the fibrotic lung drives a transcriptional and functional change in KRT5+ BCs. These findings provide insight into pulmonary epithelial cell– ECM interactions that are likely relevant to other tissues in health and disease states.

## Results

### Regional ECM organisation influences KRT5+ BC distribution

KRT5+ BCs normally line the basement membrane of the healthy human airway but have also been observed in distal lung parenchyma and HC regions in IPF[4,7]. We confirmed this in lung tissue sections from IPF patients, and normal lung tissue sections from control patients (Supplementary Fig. 1). To determine ECM organisation in regions of distal lung populated by KRT5+ BCs, we examined IPF and control lung tissue sections by second harmonic generation (SHG), 2- photon microscopy (Fig. 1a, Supplementary Table S1). SHG is a label-free, nonlinear microscopy technique that facilitates the study of collagen organisation without the need for extensive tissue processing that may disrupt ECM architecture[23,24]. This approach relies on the unique property of non-centrosymmetric structures such as fibrillar collagens including type I and III collagen – relevant to IPF - to emit a SHG signal[23]. Equivalent regions of IPF and control lung were compared, with the addition of fibrotic and HC regions that were present exclusively in IPF sections (Fig. 1b, c). In IPF lung tissue, loss of normal alveolar architecture and marked fibrotic remodelling was accompanied by the accumulation of KRT5+ BCs distributed as clusters, individual cells or lining HCs (Fig. 1c).

As expected KRT5+ BCs were highly abundant in peribronchial areas of IPF and control lungs, compared with alveolar and perivascular regions (Fig. 1d). However, in IPF, an abnormal expansion of KRT5+ BCs in fibrotic and HC regions were observed that were absent from equivalent alveolar regions in control lungs (Fig. 1d).

SHG texture analysis using a grey level cooccurrence matrix (GLCM) has been employed to study collagen organisation in conducting airways of patients with asthma[25]. In our study, quantitative analysis of SHG images for collagen density, fibre orientation and image texture (using GLCM) demonstrated significant inter-regional differences in collagen organisation (Fig. 1e–h). Principal component analysis (PCA) of quantitative image analysis data (Supplementary Table S2) showed clustering of regions based on collagen characteristics (Fig. 1e and Supplementary Fig. 2a, b). A trend towards higher collagen density was evident in the IPF peribronchial and alveolar regions compared to control which did not meet statistical significance after correction for multiple comparisons (Fig. 1f). Fibrotic and HC regions were collagen-dense and present exclusively in IPF lung tissue (Fig. 1f). IPF HC regions clustered with the peribronchial regions of IPF and controls due to increased collagen density (Fig. 1e, f) and coherency of fibre orientation (Fig. 1g) within these areas. PCA

clustering of fibrotic and perivascular regions of IPF and controls was driven by increased entropy; a measure of the degree of randomness in the image (Fig. 1h). Compared to IPF HC regions, collagen in fibrotic regions demonstrated lower inverse difference moment (IDM) and angular second moment (ASM) corresponding respectively, to lower homogeneity and lower uniformity in the images (Supplementary Fig. 2c, d). This indicates that excess collagen deposited in fibrotic regions is more disorganised than in HC regions, creating differences in matrix topography within the IPF lung. Collagen organisation in IPF HC regions was comparable to that surrounding healthy airways.

Differences in regional collagen organisation determined by SHG were correlated with KRT5+ BC number and arrangement. We found a significant positive correlation between KRT5+ BC frequency and collagen density (Fig. 1i), and fibre coherency (Fig. 1j).

Next, we used high-dimensional imaging mass cytometry (IMC) to explore ECM composition in regions of distal lung tissue adjacent to KRT5+ BCs (Fig. 2). Lung tissue sections were investigated for collagens I, III, IV, fibronectin and versican; ECM components previously identified to be enriched in fibroblastic foci[26]. In normal lung (Fig. 2a) collagen I and III – both fibrillar collagens - co-localised around airways (Fig. 2a) and blood vessels (Supplementary Fig. 3). Collagen IV - a network-forming collagen – was located at the basement membrane region of airways and alveoli (Fig. 2a) and around blood vessels (Supplementary Fig. 3). Versican - a large proteoglycan - could be visualised superficially in the airway epithelium, in interstitial areas and immune cells within the alveoli (Fig. 2 and Supplementary Fig. 3). In lung tissue from patients with IPF, collagen I, III, and IV were abundantly present adjacent to KRT5+ BCs in the peribronchial, fibrotic interstitium and HC regions (Fig. 2b–d). Versican was more abundant in the fibrotic interstitial areas and superficially within the KRT5+ epithelium in HC. Fibronectin was evident in perivascular regions in control and IPF lung tissue and was prominent in subepithelial areas of alveolar damage and fibrosis in IPF (Fig. 2c).

High-resolution microscopy of damaged alveolar regions of IPF lung showed KRT5+ BCs adjacent to alpha-SMA+ cells embedded within collagen matrix (Supplementary Fig. 4a, b) raising the possibility of direct epithelial-stromal interactions in this niche. Vimentin is a major cytoskeletal component expressed by mesenchymal cells such as fibroblasts[27], and immunostaining of the IPF lung revealed it was upregulated in the periphery of fibroblastic foci[28]. We found co-localisation of vimentin+ cells and KRT5+ cells in fibrotic alveolar regions (Supplementary Fig. 4c) and HC regions (Supplementary Fig. 4d). Crosstalk between AT2 cells and fibroblasts are important in maintaining the alveolar niche[29], however, KRT5+ BC – fibroblast interactions in the distal lung have not been previously described. Signalling between these neighbouring cells may drive pathology in the IPF lung.

### KRT5+ BC migration is directly modulated by ECM proteins

Mechanisms governing KRT5+ BC redistribution in the fibrotic lung have not been elucidated. Structural cues in the pulmonary microenvironment are critical in determining cell migration kinetics[21,30]. Given the relationship between KRT5+ BC numbers and SHG collagen characteristics, we tested the effect of different ECM constituents on KRT5+ BC migration. Human KRT5+ BCs were isolated from airway brushings and expanded in culture (Supplementary Table S3). Flow cytometry and immunofluorescence microscopy was used to validate expression of the characteristic BC markers KRT5 and p63 (Fig. 3a and Supplementary Fig. 5), and typical BC morphology was demonstrated with Airyscan (Fig. 3b). We confirmed the capacity of these KRT5+ BCs to undergo normal mucociliary differentiation in an air-liquid interface (ALI) model (Fig. 3c). KRT5+ BCs from healthy controls and IPF patients were then cultured on ECM proteins; collagen I, III, IV and versican (Fig. 3d). These were selected because of their close spatial association with KRT5+ BCs in the distal lung as confirmed by IMC (Fig. 2).

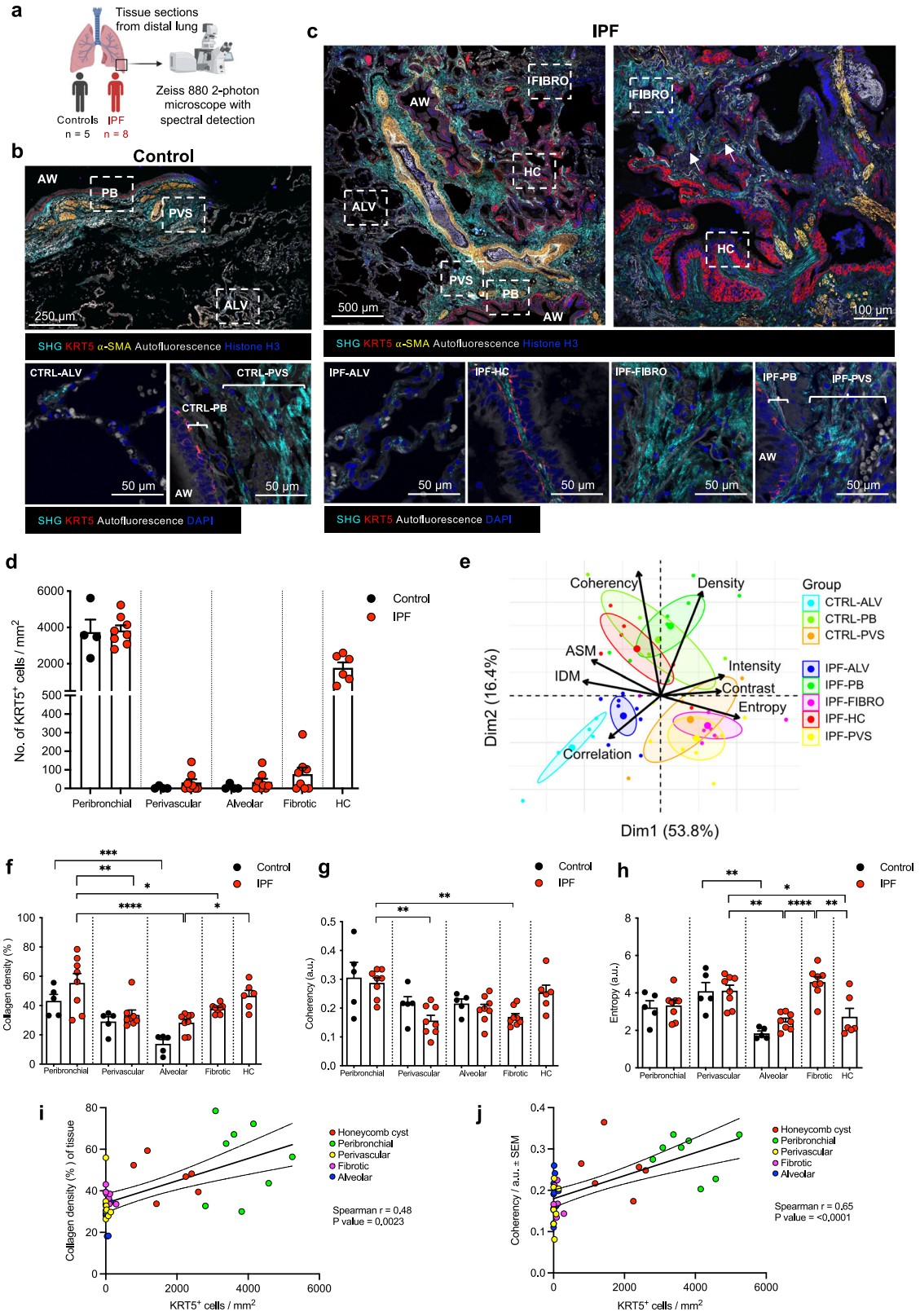

Fibronectin was not used for the cell migration assay as it promoted significant BC proliferation prohibiting accurate cell tracking analysis. We used time-lapse microscopy and semi-automated tracking to analyse cell migration over 12 h. KRT5⁺ BCs from healthy donors (*n* = 1593 cells) and IPF patients (*n* = 1574 cells) were tracked on the ECM ligands. The pattern of KRT5⁺ BC migration varied according to ECM ligand (Fig. 3e, Supplementary Movie S1). A mesenchymal-like mode of

migration was observed with exploratory lamellipodia protruding at the leading edge of cells[22]. BC mean squared displacement (MSD) over time (Fig. 3f) and cell displacement (Fig. 3g) was greater on collagen I and III compared to collagen IV and versican. Mean track speed (Fig. 3h) and straightness (or directionality) ratio (Fig. 3i) were also significantly increased on collagen I and III compared to collagen IV and versican. KRT5⁺ BCs were spatially restricted and oscillated in a

**Fig. 1 | Distinct regional collagen organisation in the fibrotic lung influences KRT5+ BC distribution. a** Lung tissue sections from control (*n* = 5) and IPF (*n* = 8) donors were imaged using a ZEISS 880 2-photon microscope with spectral detection, and linear unmixing of autofluorescence and second harmonic generation (SHG) signals. Schematic created with BioRender.com. Representative images of distal lung tissue from **b** control (*n* = 5), and **c** IPF (*n* = 8) donors with overview (upper panels) and regions quantitively analysed (lower panels). White arrows marking single KRT5+ BCs within IPF matrix in **c**. Regions included alveolar (ALV), honeycomb cyst (HC), fibrotic (FIBRO), airway (AW), peribronchial (PB) and perivascular (PVS). **d** KRT5+ BCs counted per image area; each data point represents the average of 2–6 images per area per control (*n* = 4) and IPF patient (*n* = 8 for PB, PVS, ALV, FIBRO and *n* = 6 for HC). Data plotted as mean ± SEM. **e** Principal component analysis (PCA) of collagen intensity, density, orientation and GLCM parameters for each region for controls (CTRL) vs. IPF patients. Quantitative image analysis per region showing **f** collagen density (Control PB vs. Control ALV, *P* value = 0.0007; IPF PB vs. IPF PVS, *P* value = 0.0017; IPF PB vs. IPF FIBRO, *P* value = 0.020; IPF PB vs. IPF ALV, *P* value < 0.0001; IPF ALV vs. IPF HC *P* = 0.0295) **g** coherency (IPF PB vs. IPF FIBRO, *P* value = 0.0044; IPF PB vs. IPF PVS, *P* value = 0.0012) and **h** entropy (Control PVS vs. Control ALV, *P* value = 0.0011; IPF PVS vs. IPF HC, *P* value = 0.0386; IPF PVS vs. IPF ALV, *P* value = 0.0034; IPF ALV vs. IPF FIBRO, *P* value < 0.0001, IPF FIBRO vs. IPF HC, *P* value = 0.0018). Each data point represents the average of 2–6 images per area per control (*n* = 5) and IPF patient (*n* = 8 for PB, PVS, ALV, FIBRO and *n* = 6 for HC). Data plotted as mean ± SEM. Ordinary one-way ANOVA with Tukey's multiple comparisons test ****P < 0.0001, ***P < 0.001, **P < 0.01, *P < 0.05. Spearman correlation analysis between **i** collagen density and KRT5+ BC numbers and **j** collagen coherency and KRT5+ BC numbers in distal lung tissue from IPF patients. Each data point represents the average of 2–6 images per area per IPF patient (*n* = 8 for PB, PVS, ALV, FIBRO and *n* = 6 for HC). Two-tailed *P* value reported. Linear regression of correlation analysis with best- fit line and 95% confidence bands shown. Source data are provided as a Source Data file.

fixed position on basement membrane components, collagen IV and versican (Fig. 3e). There was no significant difference comparing healthy and IPF KRT5+ BC migration on each ligand. These results show that individual ECM ligands have a profound influence on KRT5+ BC migratory dynamics.

### ECM interaction influences gene expression in KRT5+ BCs

In our 2D culture system, significant migratory differences were driven by changes in ECM ligand but not by disease-associated KRT5+ BC-intrinsic differences. Cell-derived matrix (CDM), generated by fibroblasts cultured at high density, can recapitulate the complex 3D microenvironment of tissue in simplified, in vitro models[31]. CDMs have been employed to study important biological questions, such as how normal ECM regulates cancer cell proliferation[32] but not in IPF. To investigate the effect of lung ECM on KRT5+ BC function, we established a system using human lung fibroblasts (HLFs) from IPF lung tissue (Supplementary Table S3) to generate CDMs and then studied KRT5+ BC migratory characteristics (Fig. 4a). Cultured at high-density for 21 days, IPF HLFs efficiently synthesised ECM to generate a 3D matrix (Fig. 4bi). Decellularization was performed and preservation of CDM structural integrity confirmed by immunofluorescence microscopy for ECM constituents (Fig. 4bii, biii). KRT5+ BCs from healthy controls (*n* = 213 cells) and IPF patients (*n* = 362 cells) were fluorescently labelled and tracked on the IPF CDMs over 12 h using high resolution time-lapse microscopy (Supplementary Movie S2). Broadly, two migratory patterns were observed that differed from migratory patterns observed in the 2D culture system. A subset of BCs adopted an elongated morphology with lamellipodia, migrating in a linear, directed manner according to matrix topography (Supplementary Movie S3 and S4). 20–30% of KRT5+ BCs were spatially restricted with a displacement distance of less than 20 μm and displayed an oscillatory behaviour projecting filopodia to explore the local matrix microenvironment. Cell track trajectories plotted from centroid showed overlap between healthy and IPF KRT5+ BCs (Fig. 4c). There were no significant differences in MSD over time (Fig. 4d), total displacement (Fig. 4e), track speed (Fig. 4f) or track straightness (Fig. 4g) between healthy and IPF KRT5+ BCs on IPF CDMs. This demonstrates KRT5+ BCs can migrate through a 3D fibrotic matrix and this is not influenced by cell-intrinsic disease status.

Next, we investigated how changes in ECM influence expression of genes associated with inflammation, cell migration and tissue remodelling, in KRT5+ BCs. A PCR array was performed using KRT5+ BCs from healthy controls and IPF patients cultured on collagen I (2D model) or IPF CDMs (3D model) (Fig. 4h). Discrete clustering of samples was determined by ECM microenvironment rather than KRT5+ BC disease status (Fig. 4i, j). Direct comparison of gene expression changes between healthy and IPF KRT5+ BCs on either ligand revealed very few differences (Supplementary Fig. 6). Conversely, significant gene expression changes were found in KRT5+ BCs cultured on collagen I

compared to IPF CDM (Fig. 4k). Collagen 1 alpha 2 chain (*COL1A2*) and serpin family E family member 1 (*SERPINE1*) genes were upregulated in healthy and IPF KRT5+ BCs cultured on IPF CDM. Exclusively upregulated in IPF KRT5+ cells were collagen genes (*COL3A1, COL1A1*), and MMP9, a matrix metalloproteinase. When compared to IPF CDM, KRT5+ BCs cultured on collagen I over-expressed genes encoding growth factors (*CTGF, HBEGF*), pro-inflammatory chemokines (*CXCL1, CXCL2*) and integrin subunits such as *ITGA3*. Taken together, these data indicate that the ECM microenvironment plays a critical role in determining gene expression changes in pathways relevant to tissue remodelling and fibrosis.

### IPF fibroblast- derived ECM restricts KRT5+ BC migration

To investigate whether disease-specific changes in lung ECM microenvironment differentially regulate cell migration, healthy KRT5+ BCs were tracked through control CDMs (*n* = 301 cells) compared to IPF CDMs (*n* = 257 cells) (Fig. 5a). Direct interactions between KRT5+ BCs and fibronectin fibres were demonstrated by high-resolution immunofluorescence microscopy (Fig. 5b). Visualisation of individual cell track trajectories according to CDM type (Fig. 5c) and tracks plotted from centroid (Fig. 5d), suggested reduced KRT5+ BCs migration on IPF CDMs compared to control CDMs. In accordance with this, cell MSD over time was reduced on IPF CDMs compared to control CDMs (Fig. 5e). From the gradient of the line it is possible to assess whether a particle (or cell) diffuses freely (straight line), is constrained (line plateaus) or is actively transported / trafficked (line increases gradient). As such it appears that in both groups, cells migrate in an unrestricted manner and therefore comparing displacement at the end of the experiment is an appropriate measure of migration. When KRT5+ BCs were placed onto CDM, more than half of the cells were sessile moving less than two average cell lengths (<30 μm), and the remaining cells were either motile (moving up to 90 μm) or were highly motile (moving >90 μm) (Fig. 5f). The relative proportion of motile versus highly motile cells differed significantly according to CDM type (Fig. 5f). KRT5+ BCs on control CDMs were more likely to be highly motile (Fig. 5f) and so, given this difference in proportion, sessile cells were excluded from further analysis. KRT5+ BCs showed a trend to greater displacement (Fig. 5g), track speed (Fig. 5h) and straightness ratio (Fig. 5i) on control CDMs. These findings suggested that IPF HLFs produce a matrix microenvironment that is more restrictive to KRT5+ BC migration.

### IPF fibroblast- derived ECM has a distinct matrisome

We postulated that differences in KRT5+ BC migration observed on disease compared to control matrix could be driven by compositional changes in the matrisome[33]. Previous studies have used mass spectrometry-based proteomics to characterise proteins extracted from IPF and control lung tissue resections[34–36]. In contrast, we explored the components in CDMs generated directly by IPF and

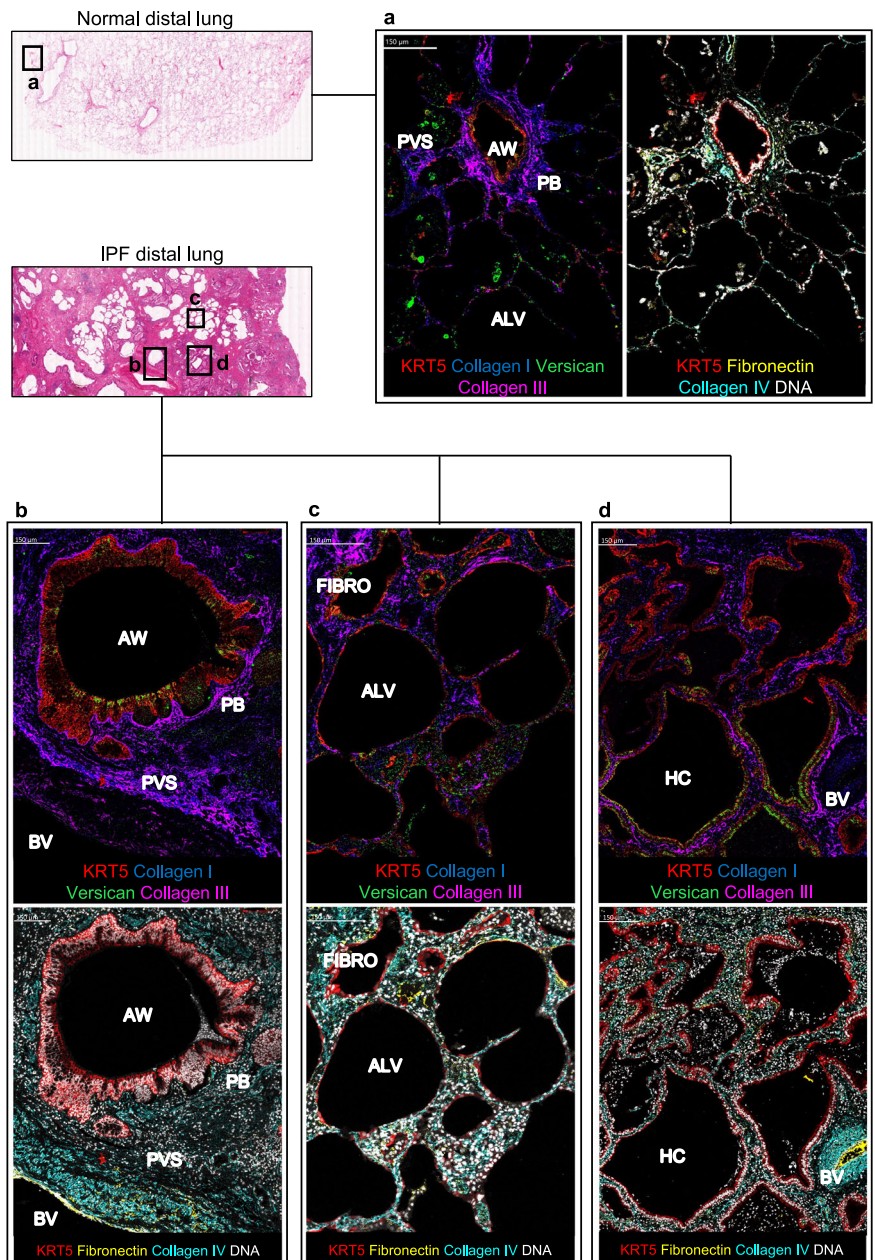

**Fig. 2 | ECM landscape of normal and fibrotic lung revealed by high-dimensional imaging mass cytometry. a** Imaging mass cytometry (IMC) of distal lung tissue from normal control ($n = 1$, H&E overview shown). Left panel; distribution of ECM components; collagen I (blue), collagen III (magenta) and versican (green). Right panel; fibronectin (yellow) and collagen IV (turquoise) in relation to KRT5⁺ BCs (red) and nuclear DNA marker (white). Representative image demonstrating key regions; airway (AW), peribronchial (PB) and alveolar (ALV). **b**–**d** IMC of distal lung tissue from a patient with IPF ($n = 1$, H&E overview shown) showing distinct regions; **b** peribronchial and perivascular space (PVS), **c** alveolar and fibrotic interstitium (FIBRO), and **d** honeycomb cyst (HC). Upper panel; collagen I (blue), collagen III (magenta), versican (green) and KRT5⁺ BCs (red). Lower panel; fibronectin (yellow), collagen IV (turquoise), KRT5⁺ BCs (red) and nuclear DNA marker (white). Scale bar, 150 μm.

control HLFs to directly identify proteins in the matrices that KRT5⁺ BCs had interacted with. Fibroblast-deposited ECM from IPF patients and non-fibrotic controls were isolated and enriched using a fractionation protocol established by Lennon et al.[37]. (Fig. 6a). 2794 proteins were identified by mass spectrometry (Supplementary Data 1). When cross-referenced with the human matrisome database[38], 156 matrix proteins were identified. Functional enrichment analysis using the Database for Annotation, Visualisation and Integrated Discovery (DAVID)[39] revealed that the top 10 GO Biological Processes included extracellular matrix organisation, cell adhesion, response to wounding and regulation of cell migration (Fig. 6b). The most abundant proteins

in the CDMs were ECM glycoproteins and regulators (Fig. 6c). PCA (Fig. 6d) and unsupervised hierarchical clustering of matrisome proteins by normalised abundance (Fig. 6e), demonstrated distinct clustering of IPF HLF derived CDMs compared to controls.

We observed 19 matrix proteins that were significantly differentially expressed between IPF and control CDMs, and of these, 14 were upregulated in IPF CDMs compared to controls (Fig. 6f). Insulin Like Growth Factor Binding Protein 5 (IGFBP-5) was the most significantly upregulated protein in IPF CDMs compared to controls (2.61 Log₂fc, $p < 0.001$) and is linked to cell migration[40]. Other significantly upregulated proteins with known roles in the regulation of cell migration

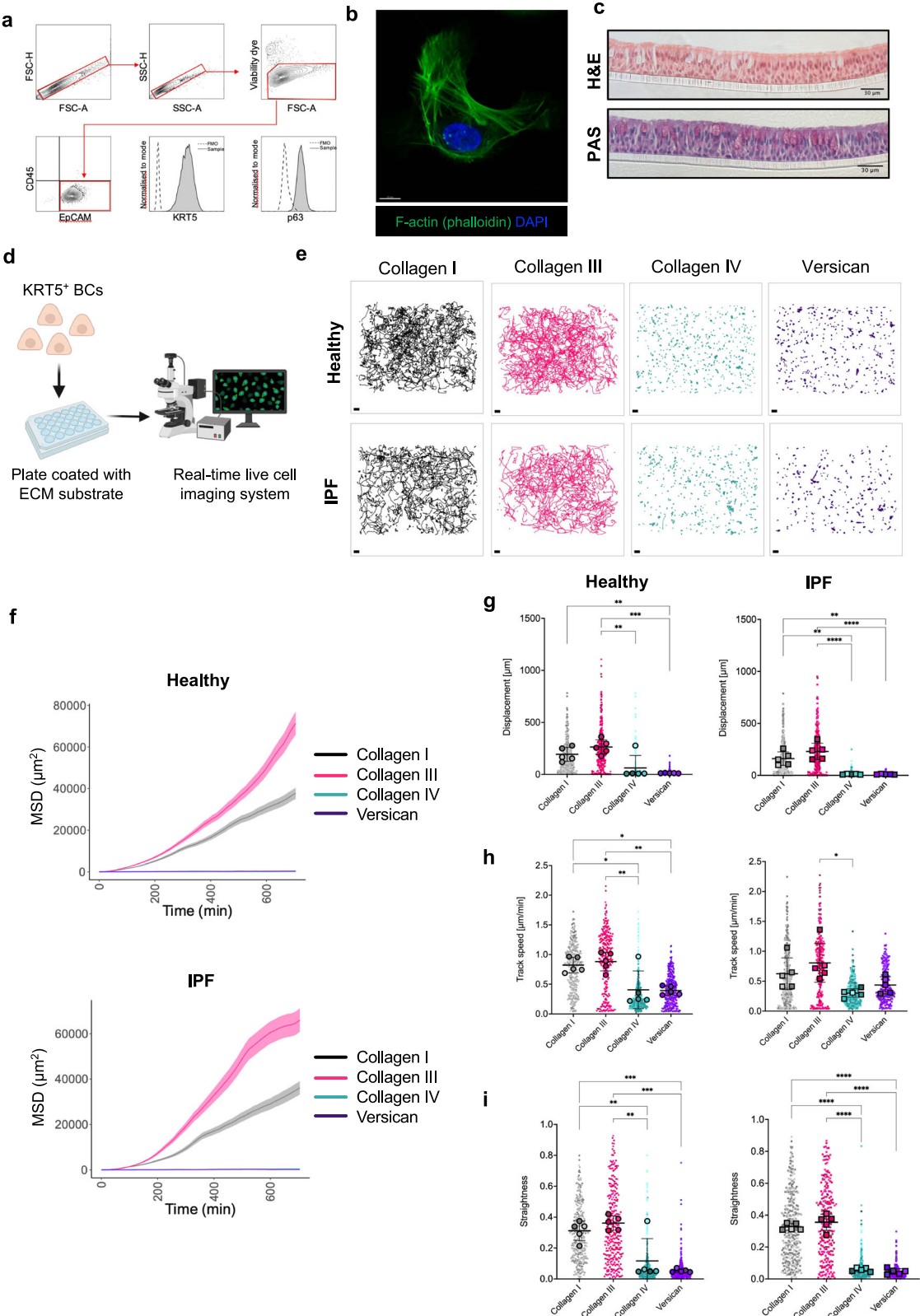

and ECM organisation, included Growth Arrest Specific Protein-6 (GAS6), Fibronectin (FN1) and Secreted Protein Acidic and Cysteine Rich (SPARC) (Fig. 6g). Lysyl oxidase (LOX) and lysyl oxidase like 2 (LOXL2) - involved in collagen cross-linking - were over-expressed in the IPF CDMs compared to controls. Serpin family F member 1 (SERPINF1) and fibrilin-1 (FBN1) – both upregulated in lung tissue from IPF patients compared to control[35] – were also significantly upregulated in

the IPF CDMs compared to control. Analysis using the STRING[41] database (https://string-db.org/) showed specific protein-protein interactions between IGFBP-5, GAS6, FN1, SPARC, Milk Fat Globule-EGF Factor 8 (MFGE8) and Fibrillin-1 (FBN1) (FDR stringency: 1%) (Fig. 6h). This analysis identifies distinct differences in the matrisome of IPF CDMs compared to controls, with many of the upregulated proteins in IPF likely to play a role in ECM organisation and cell migration.

**Fig. 3 | KRT5+ BC migration is directly modulated by ECM proteins. a** Flow cytometry confirmed expression of airway BC markers KRT5 and p63 in primary airway epithelial cells in submerged culture (passage 3). **b** Representative Airyscan maximum intensity projection image of KRT5+ BC demonstrating normal BC morphology and cytoskeletal arrangement on collagen I (*n* = 1 healthy control, *n* = 1 IPF patient); F-actin phalloidin stain (green) and DAPI (blue). Scale bar, 10 μm. **c** Normal capacity of BCs to undergo mucociliary differentiation shown here with representative H&E (**c**, upper) and PAS (**c**, lower) stained sections of agarose embedded air-liquid interface (ALI) cultures (*n* = 1 healthy control, *n* = 1 IPF patient, in duplicate). **d** Schematic of cell migration experimental design. Created with BioRender.com. **e** Individual cell track trajectories from all healthy (**e**, upper panels) and IPF (**e**, lower panels) donors plotted according to ECM ligand. Scale bars, 100 μm. **f** mean square displacement (MSD) of all KRT5+ BC tracks per ECM ligand (mean ± SEM). **g–i** Beeswarm SuperPlots showing migration data for KRT5+ BCs from healthy controls (*n* = 5) and patients with IPF (*n* = 5) cultured on collagen I (grey), collagen III (magenta), collagen IV (green) and versican (purple); **g** displacement over 12 h (healthy on collagen I vs. versican, *P* value = 0.0093; healthy on collagen III vs. versican, *P* value = 0.0005; healthy on collagen III vs.

collagen IV, *P* value = 0.0040; IPF on collagen I vs. versican, *P* value = 0.001; IPF on collagen I vs. collagen IV, *P* value = 0.0014; IPF on collagen III vs. versican, *P* value < 0.0001; IPF on collagen III vs. collagen IV, *P* value < 0.0001). **h** mean track speed (healthy on collagen I vs. versican, *P* value = 0.0132; healthy on collagen I vs. collagen IV, *P* value = 0.0164; healthy on collagen III vs. versican, *P* value = 0.0049; healthy on collagen III vs. collagen IV, *P* value = 0.0061; IPF on collagen III vs. collagen IV, *P* value = 0.0122) and **i** straightness ratio (healthy on collagen I vs. versican, *P* value = 0.0007; healthy on collagen I vs. collagen IV, *P* value = 0.0083; healthy on collagen III vs. versican, *P* value = 0.0001; healthy on collagen III vs. collagen IV, *P* value = 0.0012; IPF on collagen I vs. versican, *P* value < 0.0001; IPF on collagen I vs. collagen IV, *P* value < 0.0001, IPF on collagen III vs. versican, *P* value < 0.0001, IPF on collagen III vs. collagen IV, *P* value < 0.0001). Each small dot (healthy) or square (IPF) represents cell-level data which is colour coded according to biological replicate. The larger circles (healthy) and squares (IPF) represent the mean value per biological replicate. Summary statistics with mean and standard deviation are superimposed on the plot. Sample-level means compared using a one-way ANOVA with Tukey's multiple comparison test, ****P* < 0.0001, ***P* < 0.01, **P* < 0.05. Source data are provided as a Source Data file.

## SPARC protein modulates KRT5+ BC migration in vitro

We have shown that KRT5+ BC migration is influenced by differences in the properties of the ECM microenvironment and not cell intrinsic disease state. We next sought to functionally validate the influence of mass spectrometry-identified ECM proteins on healthy KRT5+ BC migration. The migratory dynamics of healthy KRT5+ BCs was evaluated using time-lapse imaging with recombinant IGFBP-5, GAS6, FN1 or SPARC in a 2D culture environment. For each protein we tested a range of concentrations determined by previous studies[21,40,42–46] (Supplementary Fig. 7). KRT5+ BCs do not migrate on uncoated culture plates therefore plates were pre-coated with collagen I to promote adherence and migration prior to addition of the selected ligand. It was evident that recombinant IGFBP5, GAS6 and FN1 did not appreciably alter KRT5+ BC migration on collagen I (Supplementary Fig. 7). The multifunctional ECM glycoprotein SPARC has been shown to localise to the IPF fibroblast foci by immunohistochemistry[47]. SPARC did not significantly change cell track speed (Fig. 7a) but did cause a significant reduction in straightness ratio and cell displacement in healthy KRT5+ BCs (Fig. 7b, c). When individual cell track trajectories were visualised (Fig. 7a) and plotted from centroid (Fig. 7b), the addition of SPARC to collagen I appeared to restrict KRT5+ BC migration. When quantified, SPARC caused a reduction in cell MSD over time (Fig. 7c) and total displacement (Fig. 7d). KRT5+ BC track speed (Fig. 7e) and straightness ratio (Fig. 7f) were lower with SPARC. Together these results indicate that SPARC is a unique regulator of KRT5+ BC migration. An abundance of SPARC in IPF CDMs restricts KRT5+ BC movement and its deposition by IPF HLFs represents a mechanism that could explain the retention of KRT5+ BCs in the fibrotic niche.

## Discussion

KRT5+ BCs re-populate the remodelled, fibrotic lung in advanced IPF[4,5,7,8,12], but the pathological mechanisms underlying this process have not been defined. This study provides unique insight into the interaction between KRT5+ BCs and the ECM microenvironment of the IPF distal lung. We show that KRT5+ BC migratory behaviour is modulated by ECM structure and composition. In IPF CDM compared to control CDM, KRT5+ BC movement becomes restricted. Moreover, we detected a greater abundance of the glycoprotein, SPARC in IPF CDMs and show that this molecule restricts movement of KRT5+ BCs. We postulate that through this mechanism, KRT5+ BCs are retained in fibrotic lung tissue, where they likely contribute to tissue remodelling, disease pathology and ultimately organ dysfunction.

ECM maintains tissue architecture and the biomechanical properties necessary for optimal organ function[33]. It provides signals to neighbouring cells, influencing cell phenotype and behaviour[48], and profibrotic processes in IPF[49]. While previous SHG studies[50–52] have

summarised collagen organisation in whole lung from patients with IPF, they have failed to take into account the heterogenous nature of fibrotic lung tissue. We analysed regional collagen organisation and KRT5+ BC distribution in IPF distal lung. We demonstrate dense and tangled collagen fibres in fibrotic regions of the lung – akin to those found in perivascular regions – were associated with fewer KRT5+ BCs arranged as single cells or clusters. Contrary to this collagen surrounding IPF HCs was more organised, mirroring that seen in healthy peribronchial regions, and KRT5+ BC arrangement was comparable to that found in the normal airway. Intriguingly, scRNA-seq has identified collagen-producing subpopulations residing in different anatomical locations in control and fibrotic lungs[16]. Future work is required to elucidate whether these subpopulations govern differential collagen organisation in pulmonary fibrosis.

Following lung injury in mice, distal airway progenitor cells migrate to repopulate damaged alveolar regions of the lung[12]. The complex biophysical properties of ECM including composition, topography, and stiffness determine 3D cell migration[22,53]. We have shown KRT5+ BCs possess the capacity to migrate in vitro, and this is modulated by ECM components. Previous studies of airway epithelial cell migration mainly focus on collective migration of epithelial cell sheets during wound healing[46,54–56]. In this study we sought to understand the migratory dynamics of individual KRT5+ BCs within fibrotic areas remote to the airway. In the lung, the majority of interstitial collagen is secreted and organised into sheets and cables by fibroblasts[57]. We generated CDM from HLFs to recapitulate ECM architecture and composition of native tissue[31,58]. Cell- matrix adhesions are important in modulating migration[59,60], and local microenvironmental variations in fibre stiffness in a 3D matrix altered focal adhesion dynamics and fibroblast motility[61]. It is likely that regional differences in matrix geometry within CDMs from the same donor were responsible for heterogeneity in the KRT5+ BC migratory patterns observed.

CDMs generated from IPF fibroblasts displayed a distinct matrisome and restricted KRT5+ BC migration compared to control CDMs. Many of the proteins detected in IPF CDMs have also been identified in proteomic studies of IPF whole lung tissue[34–36,62], and have been implicated in disease pathogenesis including, IGFBP-5[63], GDF-15[64], WNT5B[65], LOXL2[51], GAS6[66], FN[26] and SPARC[67,68]. IPF CDMs therefore represent a disease-relevant model with which to study epithelial – matrix interactions. SPARC is a matricellular protein secreted at significantly higher levels in conditioned media from IPF fibroblasts compared to normal lung fibroblasts[67,68]. In addition to disrupting epithelial integrity, increased SPARC promoted AT2 cell migration in a wound healing assay[68]. This effect on AT2 cell migration differs to that which we observed with KRT5+ BCs, however this may relate to

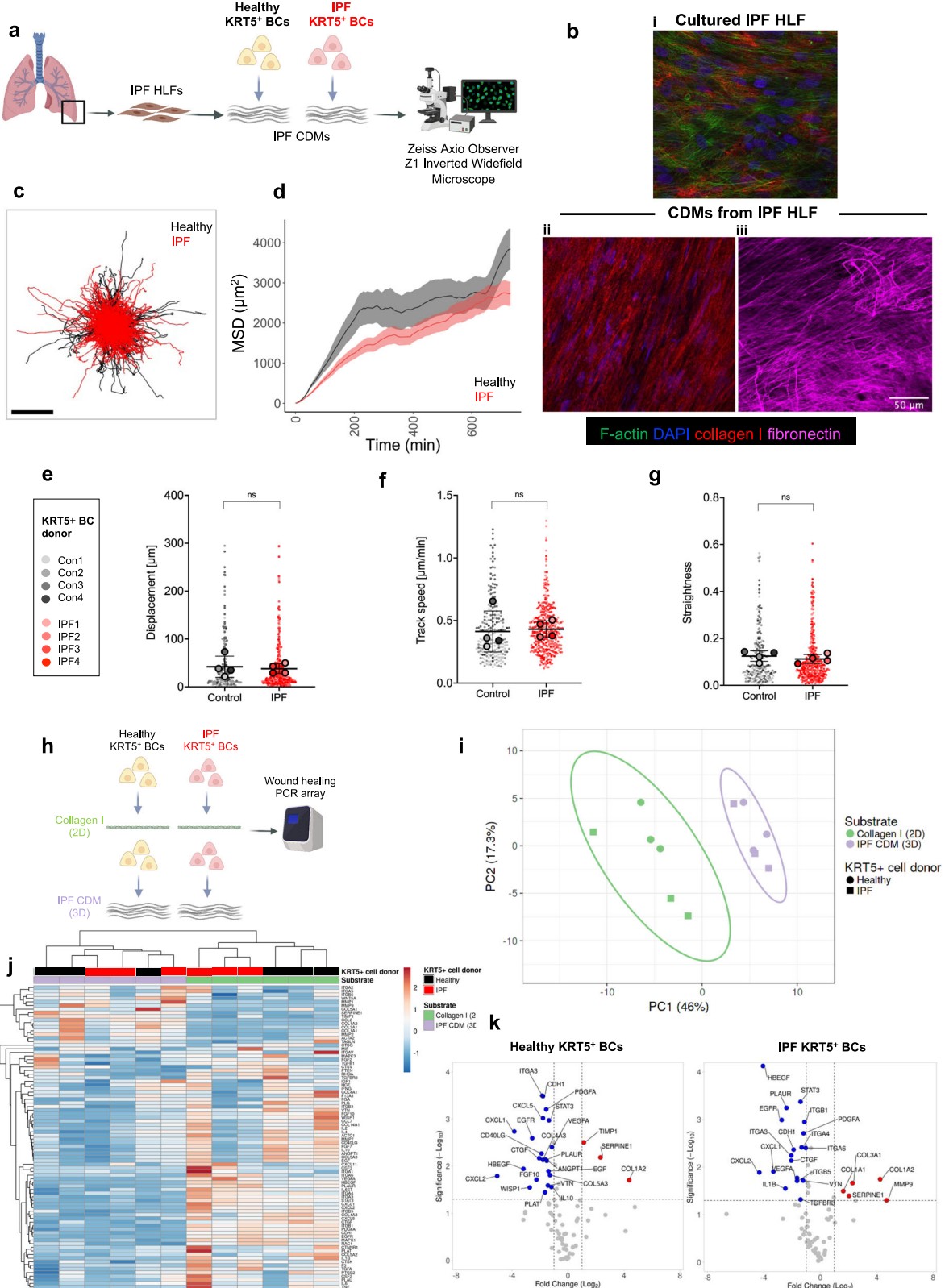

differences in migration assay and cell type studied. SPARC regulates an array of biological activities[69], and can bind fibrillar collagens and modulate collagen fibril assembly[70]. In vivo models of bleomycin-induced pulmonary fibrosis demonstrate a reduction in lung collagen content in SPARC-null mice[71] and following SPARC siRNA treatment[72]. Our data strongly support a function for SPARC in human IPF lung tissue and fibrosing lung disease. It is feasible that SPARC

overexpression in IPF ECM alters collagen assembly, influencing KRT5+ BC migration capacity.

The origin of KRT5+ BCs that accumulate in the distal IPF lung tissue is debated. Murine lineage-tracing studies have shown that following alveolar injury, KRT5+ cells arise from progenitor populations in the airway to repopulate damaged areas of lung[12,73,74]. Given species-related differences in cellular composition of the distal airways, it is

**Fig. 4 | Interaction with a 3D cell-derived matrix induces gene expression changes in migratory KRT5⁺ BCs. a** Schematic of experimental design for KRT5⁺ BC migration on CDM. Created with BioRender.com. **b** Images showing generation of CDM from primary HLFs (*n* = 1, in duplicate) **b,i** pre- and **b,ii-iii** post- decellularization; F-actin (phalloidin, green), DAPI (blue), collagen I (red) and fibronectin (purple). **c** Individual cell track trajectories from healthy *n* = 4 (black) or IPF *n* = 4 (red) KRT5⁺ BCs plotted from centroid; Scale bar, 100 μm. **d** Mean square displacement of all healthy KRT5⁺ BCs (black) vs. IPF KRT5⁺ BCs (red) on IPF CDMs (mean ± SEM). Beeswarm SuperPlot showing **e** displacement over 12 h, **f** mean track speed, and **g** straightness ratio for KRT5⁺ cells from healthy controls (*n* = 4, ≥2 replicates per donor) and IPF patients (*n* = 4, ≥2 replicates per donor) tracked over IPF HLF CDM. Each small dot represents cell-level data which is colour coded according to biological replicate. The larger circles represent the mean value per biological replicate. Summary statistics with mean and standard deviation are superimposed on the plot. Sample-level means compared using a one-way ANOVA with Tukey's multiple comparison test. **h** Schematic of experimental design to assess gene expression changes induced by culture of healthy (*n* = 3) or IPF (*n* = 3) KRT5⁺ BCs on collagen I (2D) vs. IPF CDM (3D). **i** Principal component analysis of gene array data; unit variance scaling was applied to rows; SVD with imputation was used to calculate principal components. Prediction ellipses are such that with probability 0.95, a new observation from the same group will fall inside the ellipse. **j** Heatmap with hierarchical clustering of normalised gene expression. Rows and columns using correlation distance and average linkage. Rows are centred and unit variance scaling is applied. **k** Volcano plots showing gene expression changes when healthy (left) or IPF (right) KRT5⁺ BCs were cultured on IPF CDM vs. collagen I. Upregulated genes (red), downregulated genes (blue). Significance defined as log fold change >2 or <2, *P* < 0.05. *P* values were calculated based on a Student's *t* test of the replicate normalised gene expression values (2^(- Delta CT)) for each gene in control and test groups. The *P* value calculation was based on parametric, unpaired, two-sample equal variance, two-tailed distribution. Housekeeping genes used - *ACTB, B2M* and *RPLPO*. Source data are provided as a Source Data file.

unclear how murine studies translate to human disease[75]. Single-cell transcriptional profiling revealed hypoxia and activated Notch signalling as drivers of a KRT5⁺ basal-like phenotype in human AT2 cells from fibrotic lungs[76]. Organoid and co-culture models have demonstrated the capacity of human (but not murine) AT2 cells to transdifferentiate into KRT5⁺ cells with cues from mesenchymal cells[11]. It is possible that KRT5⁺ BCs in the distal fibrotic lung are derived from more than one source.

Emerging data suggests there are molecular differences in BC subpopulations in IPF, with expansion of a 'secretory primed basal' (SPB) cell subset enriched in HCs in fibrotic lung tissue[77]. Airway basal cells from IPF patients have been shown to increase fibroblast proliferation and ECM deposition in vitro, and promote fibrotic remodelling in a humanised mouse model[78]. We did not find any disease-associated, cell- intrinsic differences in migration. Rather, we show that cues from the ECM microenvironment play a significant role in influencing the activity and mobility of KRT5⁺ BCs in the fibrotic niche.

Using human primary cells and CDMs, we developed a disease-relevant model to show that the migratory dynamics of BCs are governed by properties of the surrounding matrix. IPF HLFs produce an environment which modifies KRT5⁺ BC behaviour. Specifically, SPARC, secreted by IPF fibroblasts restricts movement of KRT5⁺ BCs and may play a key role in retaining these cells within fibrotic tissue, where they likely contribute to disease pathology and progression. Our findings have direct relevance to pathological remodelling events in IPF and may be relevant to other biological processes which involve structural cell migration within the lung.

## Methods

### Ethical approval
This research was performed in accordance with the Declaration of Helsinki. Ethical approval was granted by the Research Ethics Committee (NRES reference: 15/SC/0101, 15/LO/1399 and 15/SC/0569) at the Royal Brompton Hospital (London, UK) and all human material was collected from patients and healthy volunteers who had provided written informed consent.

### Human subjects and sample acquisition
Healthy subjects and patients with IPF were prospectively recruited at the Royal Brompton Hospital (London, UK) between March 2017 and March 2020. A diagnosis of IPF was made following multi-disciplinary discussion, according to international guidelines[79]. Healthy control subjects included previous and non-smokers with normal lung function (FEV1 > 80% predicted for age and height, FEV1/ FVC ratio >70%). All subjects underwent fibreoptic bronchoscopy in accordance with a standard operating procedure[80]. Under direct visualisation, two to three bronchial brushings (Olympus, #BC-202D-3010) were taken in the right main bronchus and placed directly into Dulbecco's Modified Eagle Medium (DMEM, Gibco).

Fresh lung tissue specimens and archived formalin fixed paraffin embedded (FFPE) lung tissue sections were obtained from the Royal Brompton Hospital (London, UK). All specimens were reviewed by a consultant histopathologist. Normal parenchymal lung tissue was obtained during resections for localised lung cancers from sites remote to any tumour mass. IPF tissue samples with evidence of a usual interstitial pneumonia (UIP) histological pattern were procured from explant lung tissue or surgical lung biopsies.

### Antibodies
Primary antibodies used for immunofluorescence were as follows: rabbit monoclonal to cytokeratin 5 AF647 (clone EP1601Y, Abcam; 1:200), rabbit polyclonal to cytokeratin 5 (clone Poly19055, BioLegend; 1:200), rabbit polyclonal anti-collagen type I (Novus Biologicals; 1:200), rabbit polyclonal anti-fibronectin (Abcam; 1:100), mouse monoclonal anti-fibronectin (clone FN-15, Sigma- Aldrich; 1:100), mouse monoclonal to alpha-smooth muscle actin Cy3 (clone 1A4, Sigma- Aldrich; 1:500), mouse anti-histone H3 (C-terminus) (clone 1B1-B2, BioLegend; 1:200). Primary antibodies used for flow cytometry were as follows: rabbit monoclonal anti-p63 (clone EPR5701, Abcam; 1:1000), mouse monoclonal anti-human CD326/ EpCAM AF488 (clone 9C4, BioLegend; 1:100), mouse monoclonal anti-human CD45 BV605 (clone H130, BioLegend; 1:100), rabbit monoclonal to cytokeratin 5 AF647 (clone EP1601Y, Abcam; 1:200). Primary antibodies used for imaging mass cytometry were as follows: rabbit monoclonal to cytokeratin 5 (clone EP1601Y, Abcam; 1:200), goat polyclonal anti-collagen type I 169Tm (Standard BioTools Inc.; 1:300), rabbit polyclonal anti-collagen type III alpha 1 (Novus Biologicals; 1:100), rabbit polyclonal anti-collagen type IV (Novus Biologicals; 1:100), recombinant rabbit monoclonal anti-versican (clone EPR12277, Abcam; 1:100), rabbit monoclonal anti-fibronectin 175Lu (clone EPR23110-46, Standard Bio-Tools Inc.; 1:100), anti-vimentin 143Nd (D21H3, Standard BioTools Inc.; 1:100).

Secondary antibodies used included: goat anti-rabbit IgG (H + L) AF 680 (ThermoFisher), goat anti-mouse IgG (H + L) DyLight 800 (Cell Signalling), goat anti-rabbit IgG DyLight 488 (Invitrogen), goat anti-rabbit IgG AF 647 (ThermoFisher), goat anti-mouse IgG AF 546 (Invitrogen), goat anti-mouse IgG AF 568 (Invitrogen).

Details of antibody supplier and catalogue number are provided in Supplementary Table S4.

### Histology and immunofluorescence microscopy
FFPE lung tissue sections (4 μm thickness) were deparaffinized by heating at 60 °C for 1 h then immersing three times for 5 min in Gentaclear (Genta Medical, UK). A graded alcohol series (100%, 90%, then 70%) followed by phosphate buffer saline (PBS) wash was used to

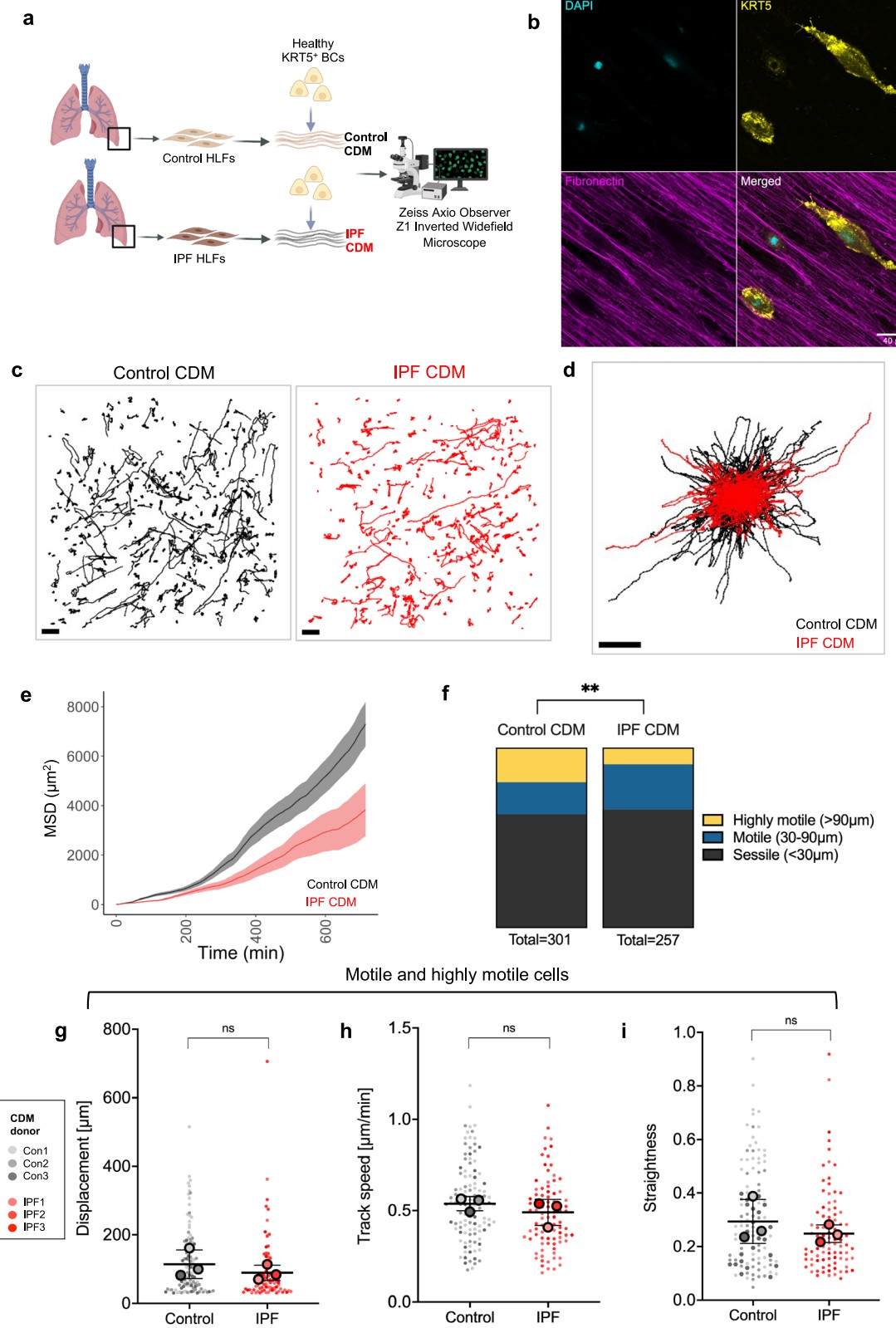

rehydrate the tissue. Antigen retrieval was performed by heating the sections in 0.01 M sodium citrate solution (pH 6.0) for 15 min at 800 watts in a microwave. To prevent non-specific primary antibody binding the sections were blocked with 10% goat serum (Sigma-Aldrich) for 1 h (room temperature). The sections were stained overnight at 4 °C with primary antibodies to KRT5 (basal cells) and alpha-smooth muscle actin (myofibroblasts). After washing in PBS,

secondary antibodies diluted in 10% goat serum were added for 30 min at room temperature. The secondary antibody used was goat anti-rabbit IgG conjugated to DyLight® 488 (1:200; Invitrogen). Nuclei were counterstained with DAPI (Sigma-Aldrich #D9542; 1:1000) or anti-Histone H3 (C-terminus) (BioLegend; 1:200) for 20 min. ProLong Gold antifade mountant (Invitrogen; #P36934) was applied directly to the tissue sections. As a negative control, tissue stained only with the

**Fig. 5 | IPF fibroblast-derived matrix restricts KRT5⁺ BC migration. a** Schematic of experimental design for KRT5⁺ BC migration on CDM from control or IPF HLFs. Created with BioRender.com. **b** Images showing KRT5⁺ BC interacting with fibronectin fibres; KRT5 (yellow), DAPI (turquoise) and fibronectin (purple). **c** Individual cell track trajectories for all healthy KRT5⁺ cells on $n = 3$ control CDMs (black) or $n = 3$ IPF CDMs (red); Scale bar, 100 μm. **d** Cell track trajectories plotted from centroid; Scale bar, 100 μm. **e** Mean square displacement (MSD) over time of all healthy KRT5⁺ BCs on $n = 3$ control (black) on $n = 3$ IPF (red) CDMs (mean ± SEM). **f** Contingency analysis of cell proportions according to displacement distance for CDM type, **$P$ value = 0.0013, calculated using a Chi-squared test. **g**–**i** Beeswarm

SuperPlots showing **g** displacement over 12 h, **h** mean track speed, and **i** straightness ratio for motile and highly motile KRT5⁺ BCs from healthy controls tracked over CDMs from control ($n = 3$, ≥2 replicates per donor) or IPF HLFs ($n = 3$, ≥2 replicates per donor). Each small dot represents cell-level data which is colour coded according to biological replicate. The larger circles represent the mean value per biological replicate. Summary statistics with mean and standard deviation are superimposed on the plot. Sample-level means compared using a one-way ANOVA with Tukey's multiple comparison test. Source data are provided as a Source Data file.

secondary antibody was used. The sections were viewed on a Leica SP5 inverted confocal microscope using Leica LAS software. Images were acquired on a Leica SP5 inverted confocal microscope equipped with four lasers (405 nm, 488 nm, 543 nm, 633 nm), using a 10×, 20× and 63× oil objective lens, and Leica Application Suite X (LAS X) software. Images were processed in Fiji v1.52p software[81]. Regional enumeration of fluorescently labelled KRT5⁺ cells associated with collagen matrix was performed manually in Fiji v1.52p (http://fiji.sc).

### Mutiplexed imaging

Multiplexed imaging of distal lung was adapted from McCowan et al.[82]. Antigen retrieval was performed on histological sections using Target Retrieval Solution, pH 6.0 (Agilent) (20 min at 97 °C) on a Dako PT retrieval module. Samples were permeabilized and blocked for 20 min in PBS/Neutral goat serum (NGS) 10%/BSA1%/TritonX-100 (Tx100) 0.3%/Azide 0.05% at 37 °C and stained with 150 μl rabbit anti-Keratin 5 (Polyclonal, Poly19055, BioLegend; 1:200) diluted in PBS/ NGS10%/ BSA1%/ TX-100 0.3%/Azide 0.05% for 1 h. Samples were washed 3 times with PBS/BSA1%/TX-100 0.1%/Azide 0.05% before adding 150 μl of a solution containing, anti-Histone H3 (C-terminus) AF 594 (BioLegend; 1:200), aSMA-Cy3 (clone 1A4, Sigma; 1:1000) and anti-rabbit-AF647 (polyclonal, A-21244, ThermoFisher) diluted in PBS/ NGS10%/BSA1%/ TX-100 0.3%/Azide 0.05% for 1 h. Samples were washed 3 times with PBS/BSA1%/TX-100 0.1%/Azide 0.05% and 2 times in PBS. Finally slides were mounted with Vectashield (Vector Laboratories, #H-1000-10).

### Second harmonic generation imaging

To obtain concurrent second harmonic generation (SHG) images, stained tissue sections were also imaged a ZEISS LSM 880 NLO multiphoton microscope (Beatson Advanced Imaging Resources, Beatson Institute) equipped with a 32 channel Gallium arsenide phosphide (GaAsP) spectral detector (ZEISS) using 20×/1 NA water immersion objective lens. Samples were excited with a tuneable multiphoton laser (680–1300 nm) set up at 900 nm or 100 nm and/or normal lasers (488 nm, 561 nm and 633 nm) depending on immunostaining used, and signal was collected onto a linear array of the 32 GaAsp detectors in lambda mode with a resolution of 8.9 nm over the visible spectrum. Spectral images were then unmixed with ZEISS ZEN software using references spectra acquired from unstained tissues (tissue autofluorescence and Second Harmonic Generation), DAPI-stained tissue or slides labelled with AF488-, Cy3-, AF594- or AF647-conjugated antibodies. Images were acquired in each of the following regions: alveolar (ALV-NORM, ALV-IPF), airway peribronchial (PB-NORM, PB-IPF), perivascular space (PVS-NORM, PVS-IPF), honeycomb cyst (CYST) and fibrotic (FIBRO).

### Image texture analysis

To analyse the structural characteristics of collagen fibres in each region, a variety of measurements were obtained using Imaris v9.6.0 (Oxford Instruments) and OrientationJ[83] (http://bigwww.epfl.ch/demo/orientation/) and GLCM (https://imageJ.nih.gov/ij/plugins/texture.html) plugins for (Fiji/ ImageJ). Imaris v9.6.0 (Oxford Instruments) surface tool was used to segment collagen fibres and tissue autofluorescence to measure collagen intensity and density (defined as % of

tissue covered with collagen), and to quantify the number of KRT5⁺ cells in the different analysed regions. OrientationJ was used to measure fibre coherency utilising properties of the original image, so-called 'first-order statistics'. Grey-level co-occurrence matrix (GLCM) texture analysis derives statistical measures by considering the spatial relationship of pixels in an image, so-called 'second-order statistics'. The GLCM features measured included contrast, entropy, correlation, inverse difference moment (IDM) and angular second moment (ASM) (Supplementary Table S2).

### Imaging mass cytometry (IMC)

**Antibody panel design.** An antibody panel was designed to capture ECM microenvironment surrounding KRT5⁺ BCs in the distal lung. Pre-chelated antibodies were used to stain for collagen type I – 169Tm (Standard BioTools Inc., #3169023D, 1:300), fibronectin – 175Lu (Standard BioTools Inc.; 1:100) and vimentin – 143Nd (Standard BioTools Inc., #3143027D, 1:100). A Maxpar X8 antibody labelling kit (Standard BioTools Inc.; #201151B) was used to chelate 100 μg of BSA- and azide- free, purified antibody as per the manufacturer's protocol; anti- cytokeratin 5 (clone EP1601Y, Abcam; 1:200) was conjugated with the 164Dy metal isotope (Standard BioTools Inc., #201164A), anti-collagen III (Novus Biologicals; 1:100) with the 176Yb isotope (Standard Biotools Inc., #201176A), anti-collagen IV (Novus Biologicals; 1:100) with the 153Eu isotope (Standard BioTools Inc., #201153A), and anti-versican (clone EPR12277, Abcam; 1:100) with the 147Sm isotope (Standard BioTools Inc., #201147A). Each custom - chelated antibody was tested on FFPE tissue sections from controls and patients with IPF to ensure an expected staining pattern and optimal dilution with good signal-to-noise ratio for each channel. The dilution factor was altered for channels with obvious spillover into neighbouring channels. Nuclear staining was conducted using the Cell-ID™ Intercalator kit (191Ir and 193Ir, Standard BioTools Inc, #201192A).

**Imaging.** FFPE lung tissue sections (4 μm thickness) from controls and patients with IPF were stained with H&E and imaged on Aperio Versa Slide Scanner (Leica Biosystems). Tissue integrity and suitability for IMC was assessed in Aperio ImageScope (Leica Biosystems). Suitable serial FFPE lung tissue sections were dewaxed by baking in a dry oven at 60 °C for 2 h. The sections were placed in Gentaclear (Genta Medical) for 10 min, followed by rehydration in a graded series of ethanol (100%, 95%, 80%, 70%) for 5 min each. Sections were washed on a shaker in Maxpar water (Standard BioTools Inc.) for 5 min. Heat-induced antigen retrieval was performed at 96 °C in Dako Target Retrieval Solution (Agilent, #S-236784-2; diluted from 10x to 1x). After cooling to less than 70 °C, sections were washed with Maxpar water and Maxpar phosphate-buffered saline (PBS; Standard BioTools Inc). The sections were blocked with 3% Bovine serum albumin (BSA; Sigma-Aldrich) in Maxpar PBS (Standard BioTools Inc) for 45 min at room temperature (RT). Sections were incubated at 4 °C overnight with primary antibodies in 0.5% BSA (see resources section). Slides were washed x2 for 8 min in 0.2% Triton X-100 (Sigma-Aldrich) in Maxpar PBS (Standard BioTools Inc), followed by 2 washes in Maxpar PBS (8 min per wash). Sections were counterstained with Cell-ID™ Intercalator-Ir (Standard BioTools Inc., #201192A; 1:400) added to

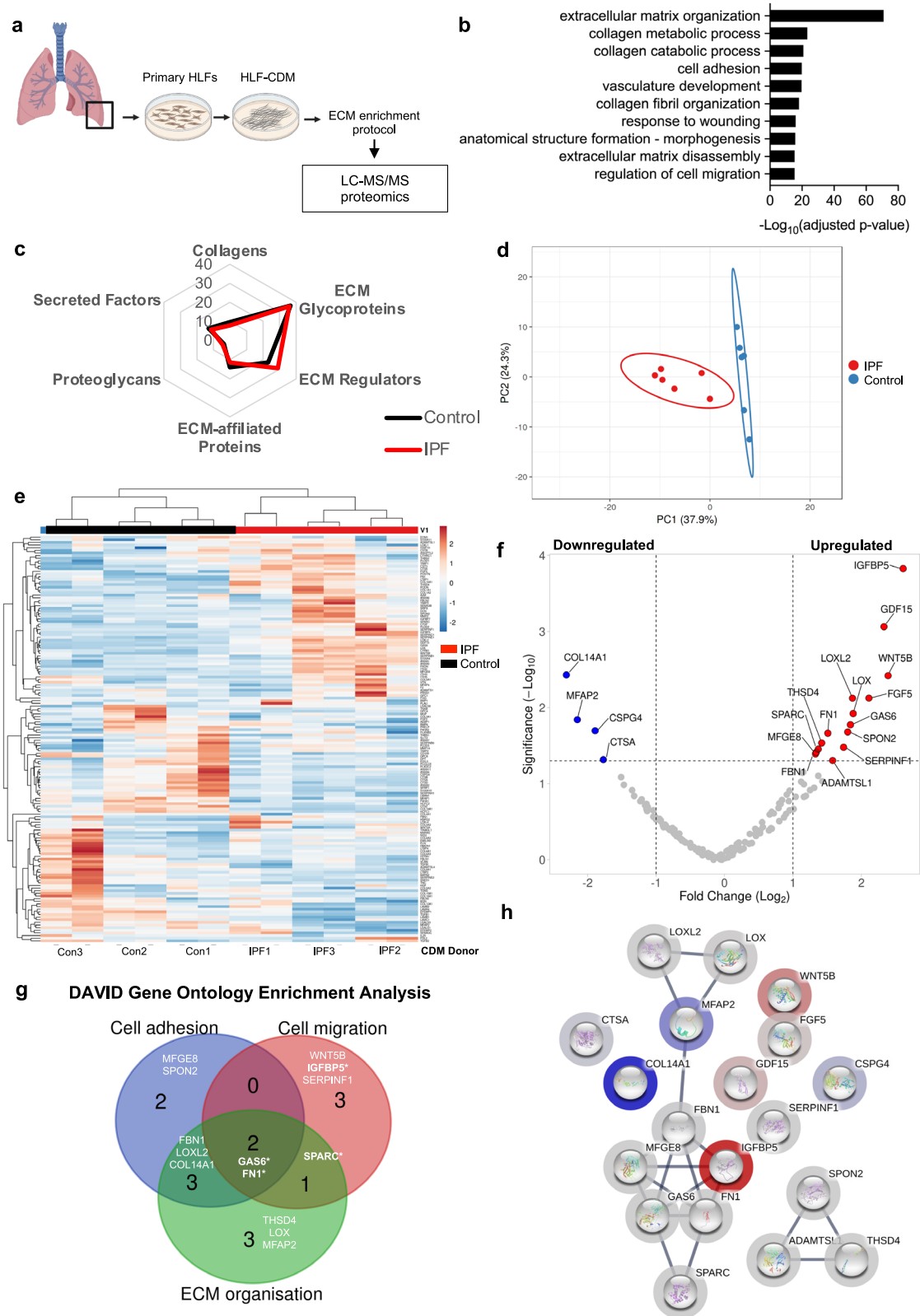

each section and incubated at RT for 30 min. Sections were washed in Maxpar water for 5 min and air-dried for 30 min. The slides were stored without coverslips at RT until imaging. The Hyperion Imaging System (Standard BioTools Inc) was used to attain IMC images. Following start-up in CyTOF Software v7.0 (Standard BioTools Inc), the system was calibrated using a 3-element full coverage tuning slide (Standard Bio-Tools Inc) which utilises a polymer matrix containing Yttrium-89,

Cerium-140 and Lutetium-175. On completion of successful calibration, simple brightfield images of the tissue were taken using the 'Panorama' function, Regions of interest (ROIs) were selected within areas of AR. Laser ablation was performed with laser power 2, frequency 200 Hz and 1 μm resolution. Raw data were exported from CyTOF 7.0 in.mcd format. MCD Viewer v1.0.560.6 (Standard BioTools Inc) was used to review each channel to ascertain image quality and

**Fig. 6 | IPF fibroblast-derived matrix has a distinct matrisome. a** Schematic of experimental design for MS-proteomics of CDMs from control ($n = 3$ in duplicate) and IPF ($n = 3$ in duplicate) HLFs. Created with BioRender.com. **b** GO Biological Pathways using DAVID v6.8 functional annotation tool for 154 matrisome proteins identified in CDMs (one-sided Fisher's exact test, Benjamini–Hochberg adjusted $P$ value). **c** Radar plot of protein abundance according to matrisome categories identified in control (black) and IPF (red) CDMs. **d** PCA of 154 matrisome proteins identified in control and IPF CDMs. Unit variance scaling is applied to rows; SVD with imputation is used to calculate principal components. Prediction ellipses are such that with probability 0.95, a new observation from the same group will fall inside the ellipse. **e** Heatmap with hierarchical clustering of normalised abundance of each protein identified. Rows and columns using correlation distance and average linkage. Rows are centred and unit variance scaling is applied. **f** Volcano plot showing significant differentially expressed proteins between IPF CDMs and control CDMs, upregulated proteins (red), downregulated proteins (blue). One-way ANOVA, Benjamini–Hochberg adjusted $P$ value, significance defined as a fold change >2 or <2, $P < 0.05$. **g** Venn diagram showing shared GO biological processes for significantly differentially expressed proteins. **h** Protein-protein interaction network using STRING[41] v11 using matrisome proteins with an abundance ratio >2 or <2, $P < 0.05$. Source data are provided as a Source Data file.

pseudo-colours were assigned to channels of interest to generate a composite image which was exported as a.tiff file.

## Isolation and culture of primary human bronchial epithelial cells (PBECs)

**Submerged culture.** Freshly collected bronchial brushings were agitated in media to detach cells and centrifuged ($300 \times g$, 5 min). The resultant cell pellet was resuspended in serum-free, Airway Epithelial Growth Medium (PromoCell) containing manufacturer's supplements and Primocin (InvivoGen, #ant-pm-1), added to protect against microbial contamination. The cell suspension was seeded in a 25 cm² flask (Corning) pre-coated with purified type 1 bovine collagen (Pure-Col, Advanced BioMatrix, #5005). Cells were cultivated at 37 °C in a humidified atmosphere of 5% CO$_2$ and media was refreshed every 48-h. PBECs were passaged at 80% confluence and cryopreserved in PromoCell media with 10% DMSO (Sigma-Aldrich) and 10% FBS (Gibco) (passage #2). When required, PBECs were thawed quickly in a 37 °C water bath and cultured in a pre-coated 75 cm² flask (Corning) until 80% confluent, then passaged for use in experiments (Supplementary Table S3).

**Air-liquid interface culture.** Passage #3 KRT5$^+$ cells were seeded (4.0 ×10⁴/ insert) on purified collagen 1 (Advanced BioMatrix) coated Transwell polyester membrane inserts (6.5 mm diameter; 0.4 μm pore size; Corning). PneumoCult-Ex media (STEMCELL) containing manufacturer's supplements, was added to apical and basal chambers and replaced 48-hourly. When confluent, the apical media was removed to promote mucociliary differentiation, and PneumoCult-ALI medium with manufactuer's supplements (STEMCELL) and Primocin (InvivoGen, #ant-pm-1) was added to the basal chamber. From week 2 following air- exposure, mucus was gently washed from the apical surface with warmed PBS and stored for further analysis. Cells were maintained in ALI culture for 21 days, then inserts were fixed in 4% PFA (Electron Microscopy Sciences, #RT-15710) for immunofluorescence microscopy.

## Flow cytometry of cultured PBECs

Cultured primary airway epithelial cells (passage #3) were plated in a 96-well round bottomed plate (2 ×10⁵/ well). Cells were washed in PBS (Gibco), centrifuged (500 × g, 4 °C, 5 min) and the pellet resuspended in LIVE/DEAD fixable near-IR dead cell stain (Invitrogen, #L10119; 1:1000). After a 20-min incubation in the dark at room temperature, excess viability dye was removed with a PBS wash. Intracellular fixation and permeabilization buffer (eBioscience; #88-8824-00) was added to the cells overnight at 4 °C in the dark. Cells were washed in PBS and stained with 150 μl unconjugated primary antibody anti-p63 and Human Fc block (BD Pharmingen, #564219; 1:50) diluted in permeabilization buffer (eBioscience) for 30 min at 4 °C in the dark. Cells were washed in permeabilization buffer and centrifuged. Secondary antibody staining was performed with Goat Anti-Rabbit IgG conjugated to BV421 (1:200; BD) diluted with permeabilization buffer in 2% BSA. Cells were washed and stained with remaining conjugated primary antibodies to EpCAM/ CD326, CD45 and KRT5 diluted in permeabilization buffer for 30 min at 4 °C in the dark. After a final washing step, the cells were resuspended in FACS buffer. Flow cytometry was performed using a BD LSR Fortessa III cell analyser. FlowJo v7 was used for data analysis.

## Isolation and culture of primary human lung fibroblasts (HLFs)

HLF were extracted from lung tissue specimens using an established protocol[84]. Linear, parallel scratches were made with a scalpel on the surface of a 6- well culture plate to encourage fibroblast outgrowth. Lung tissue was cut into 1 mm³ fragments and evenly distributed within each well (5 pieces/well). HLFs were observed growing out of the tissue fragments after approximately 7 days in culture. Explant cultures were maintained in DMEM (supplemented with 20% FBS and Primocin) in an incubator (37 °C, 5% CO$_2$). Media was refreshed every 48 h. Primary HLFs were passaged at 80% confluence and cryopreserved in DMEM with 10% DMSO and 10% FBS (passage #2). When required, HLFs were thawed quickly in a 37 °C water bath and cultured in a 75 cm² flask (Corning) until 80% confluent, then passaged for use in experiments (Supplementary Table S3).

## Cell-derived matrices (CDMs) from primary HLFs

**Generation of CDMs.** The approach used to generate CDMs from primary HLFs was based on a protocol published by Kaukonen et al.[31]. HLFs isolated from IPF ($n = 3$) and control ($n = 3$) subjects were seeded ($4.0 \times 10^4$) in wells of a 24-well μ-plate (Ibidi). The cultures were maintained in DMEM (Gibco) media supplemented with 10% FBS (Sigma-Aldrich) and Primocin (InvivoGen, #ant-pm-1) in an incubator (37 °C, 5% CO$_2$). The cells formed a confluent monolayer and media was replaced every 48 h for 21 days to allow CDM production by HLFs. Medium was removed and the cells washed with PBS. HLFs were denuded by adding 500 μl warmed sterile- extraction solution (1 ml of NH$_4$OH, 250 μl of Triton-X-100 and 48.75 ml PBS passed through a 0.45 μm filter) for approximately 1 min whilst closely observing the integrity of the matrix using a light microscope. The extraction buffer was gently removed and the CDM washed carefully with PBS to avoid disruption. Residual cellular DNA was removed by incubating CDMs (37 °C, 30 min) with 500 μl DNase I (Roche, #10104159001; 10 μg ml⁻¹). The CDM was gently washed with PBS and matrix integrity was validated using a light-microscope. In contrast to the protocol described by Kaukonen et al.[31]. we did not use gelatin-coated coverslips or add ascorbic acid to the media. Ascorbic acid plays an important role in secretion and assembly of human collagens[85].This study was not aimed at collagens specifically but rather the production of multiple matrisome proteins and the way these influenced cell migration, irrespective of modifications introduced by ascorbic acid supplementation. CDMs from control and IPF fibroblasts were generated under the same conditions.

**Immunofluorescence microscopy.** CDMs in a 24-well μ-plates (Ibidi) were gently washed in PBS and fixed in 10% formalin (Sigma-Aldrich, #HT5011-ICS) for 15 min at room temperature. The fixative was removed, CDMs were washed in PBS then blocked with 30% horse serum (Gibco) for 1 h at room temperature. CDMs were stained overnight at 4 °C with primary antibodies to ECM constituent's collagen I and fibronectin. After careful washing in PBS, secondary antibodies diluted in 30% horse serum were added for 30 min at room

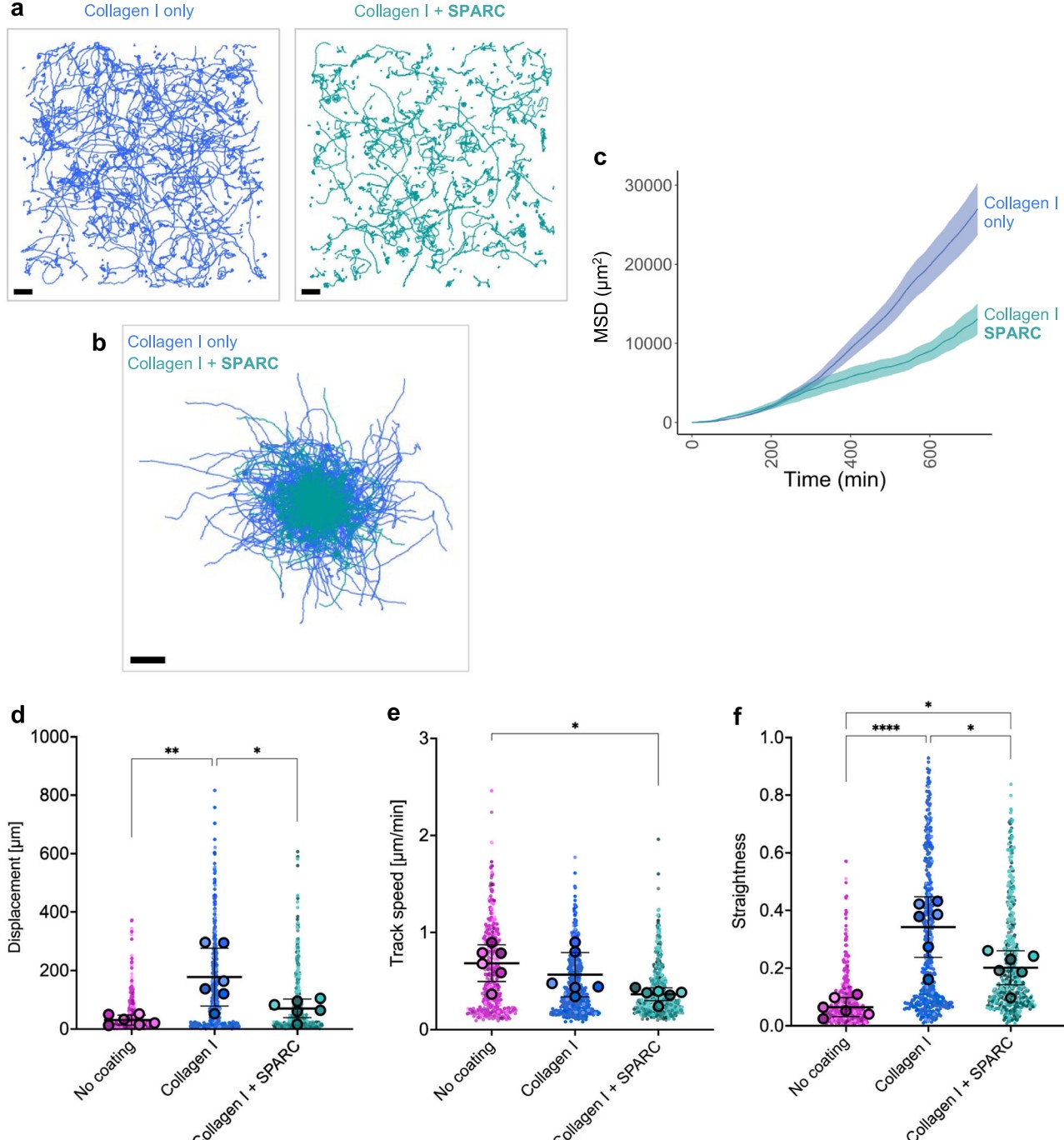

**Fig. 7 | SPARC restricts KRT5+ BC migration. a** Individual cell track trajectories for healthy KRT5+ BCs on collagen I (blue) or collagen I and SPARC (green), Scale bar, 100 μm **b** Cell tracks plotted from centroid; Scale bar, 100 μm. **c** Mean square displacement (MSD) over time of all healthy KRT5+ BCs on collagen I (blue) or collagen I and SPARC (green) (mean ± SEM). **d**–**f** Beeswarm SuperPlots showing migration data for KRT5+ BCs from healthy controls (n = 6, in triplicate) cultured in wells with no coating (magenta), collagen I (blue) or collagen I plus SPARC 20 μg/ml (green); **d** displacement over 12 h (BCs on no coating vs. collagen I, P value = 0.0021; BCs on collagen I vs. collagen I plus SPARC, P value = 0.0205), **e** mean track speed (BCs on no coating vs. collagen I plus SPARC, P value = 0.0166), and **f** straightness ratio (BCs on no coating vs. collagen I, P value < 0.0001; BCs on no coating vs. collagen I plus SPARC, P value = 0.0132; BCs on collagen I vs. collagen I plus SPARC, P value = 0.0108). Each small dot represents cell-level data which is colour coded according to biological replicate. The larger circles represent the mean value per biological replicate. Summary statistics with mean and standard deviation are superimposed on the plot. Sample-level means compared using a one-way ANOVA with Tukey's multiple comparison test, ****P < 0.0001, **P < 0.01, *P < 0.05. Source data are provided as a Source Data file.

temperature. To validate the decellularization procedure, the secondary antibody used was goat anti-rabbit IgG conjugated to Alexa Fluor 546 (1:200; Invitrogen). For the cell migration experiments, the secondary antibody used was goat anti-mouse Alexa fluor 568 (1:200; Invitrogen). In both experiments Phalloidin-iFluor 488 (Abcam, #ab176753) was used as per manufacturer's instructions and nuclei

were counterstained with DAPI (Sigma-Aldrich #D9542; 1:1000) for 20 min. The staining solution was removed from each well and Vectashield Antifade Mounting Medium (Vector Laboratories, #H-1000-10) was added. Imaging was carried out using a 63x oil objective on a Leica SP5 inverted confocal microscope. Images were acquired using the LAS AF platform (Leica) and analysed using Fiji v1.52p software[81].

The KRT5$^+$ cell shown (yellow) migrating on fibronectin (magenta) in Fig. 5b, was imaged on a Leica SP8 confocal, with LAS X software, using a 20x/0.75NA Plan Apo lens. The figure was created in Icy[86], an open source bioimage analysis software (https://icy.bioimageanalysis.org/).

### KRT5$^+$ cell migration experiments

**KRT5$^+$ cell migration on ECM ligands.** Tissue culture plates (48-well, Corning) were coated with purified human collagen type I (Sigma-Aldrich, #CC050), type III (Sigma-Aldrich, #CC054), type IV (Sigma-Aldrich, #C6745) or recombinant human versican (RayBiotech, #230-00833-50) at a concentration of 50 µg/ml, for 1 h (37 °C, 5% CO$_2$) in line with previously published methods from our lab[21]. KRT5$^+$ cells from healthy controls and IPF patients were seeded (2.5 ×10$^3$) into each well and after 30 min to adhere, were imaged with a 4x objective on the JuLI™ Stage Real-Time Cell History Recorder every 15 min for 48 h. The JPEG images generated were imported to Nikon NIS Elements v.4.50 for analysis.

**KRT5$^+$ cell migration through CDMs.** KRT5$^+$ cells were incubated with CellTracker™ Deep Red Dye staining solution (Invitrogen, #C34565) for 40 min at 37 °C. The cells were centrifuged (1200 rpm, 5 min, 4 °C), staining dye removed and resuspended in Airway Epithelial Cell Growth media (PromoCell). KRT5$^+$ cells were added (1.0 × 10$^4$ cells / well) to decellurised CDMs in a 24-well µ-plate (Ibidi). The cells were incubated for 2 h (37 °C, 5% CO$_2$) to allow the cells to adhere to the CDM. A ZEISS Axio Observer Z1 Inverted Widefield Microscope with Lumencor SpectraX LED illumination and ZEISS ZEN (Blue Edition) acquisition software was used to image the fluorescently labelled cells. After plate calibration, Köhler illumination was adjusted, and positions in each well were added manually. Images were captured in the far-red channel (excitation 625−655 nm, emission 665−715 nm, for CellTracker DeepRed, #C34565) and in transmitted light, using a Plan-Neofluar 10x/0.30 Ph1 objective. The system had a Hamamatsu Flash4.0 camera, and 2 × 2 binning was used, giving a 1024×1024 pixel image with a pixel size of 1.30 µm. Two positions with good cell coverage were manually selected per well, and their x, y and z coordinates saved. A Z-stack of five slices with a step of 7 µm was acquired for each position, covering 28 µm in total. The stacks were centred on the z value saved when selecting the position, and these Z-stacks were acquired every 5 min for 24 h. To minimise subtle displacement of the delicate matrix in the well associated with stage-related turbulence in the culture media, the stage speed was reduced to 40%. At the end of the experiment the CDMs with adherent KRT5$^+$ cells were fixed in 10% formalin (Sigma-Aldrich, #HT5011-ICS) for 15 min at room temperature, washed in PBS and stored at 4 °C for subsequent immunofluorescent imaging.

**KRT5$^+$ cell migration on ECM proteins identified through mass spectrometry-proteomics.** Tissue culture plates (48-well, Corning) were coated with purified type 1 bovine collagen (PureCol, Advanced BioMatrix; 1:100) for 1 h in an incubator. The following ECM proteins were added to each well at a range of concentrations determined by previous studies:[21,40,42−46] Recombinant Human SPARC Protein (R&D Systems, #941-SP-050; 1 µg/ml, 10 µg/ml, 20 µg/ml), Recombinant Human IGFBP-5 Protein (R&D Systems, #875-B5-025; 0.2 µg/ml, 0.5 µg/ml, 1.0 µg/ml), Recombinant Human GAS6 Protein (R&D Systems, #885-GSB-050; 0.1 µg/ml, 0.4 µg/ml, 1.0 µg/ml) and Human Plasma Fibronectin Purified Protein (Sigma-Aldrich, #FC010; 10 µg/ml, 50 µg/ml, 100 µg/ml). The plate was incubated for 2 h (37 °C, 5% CO$_2$). Wells that were uncoated and only coated with type 1 collagen were used as controls. Healthy KRT5$^+$ cells (2.5 ×10$^3$) were added to each well and incubated for 2 h (37 °C, 5% CO$_2$) to allow the cells to adhere. A ZEISS Axio Observer Z1 Inverted Widefield Microscope with Lumencor SpectraX LED illumination and ZEISS ZEN (Blue Edition) acquisition

software was used to image the cells. Microscopy setup as previously described.

### Cell tracking analysis

**KRT5$^+$ cell migration on ECM ligands.** The JPEG Images generated by the JuLI real-time cell history recorder were imported to Nikon NIS Elements v.4.50 (Nikon Instruments Inc.) to create an image sequence and calibrated (1.13 µm/ pixel in x and y). The sequence was then saved as a.nd2 file for processing and analysis. To remove any artefact associated with microscope drift, the sequence was aligned to the previous frame. A median filter with a kernel size of 9 was applied to improve accuracy of spot detection. Spot detection using the brightfield channel was set to detect dark spots with a typical diameter of 27 µm and with a contrast setting of 8. Bright objects were removed. Once dark spots were detected, automated tracking was performed by the software algorithm 'Track Binaries' and then manually inspected as a means of quality control. False tracks such as those associated with debris or those which jumped to nearby cells were removed from the analysis. If overall tracking was deemed less than 90% accurate on the basis of manually checking the tracks, the sequence was excluded. The raw cell migration data generated by Nikon NIS software was exported to MotilityLab (http://www.motilitylab.net/) an online resource allowing the quantitative and statistical analyses of cell tracks, and Graph-Pad Prism v9.4.1 for further analysis and presentation. Only cells tracked for the entire 12-h duration were included in the final analysis. For each cell the following measurements were reported; (1) track speed - track length divided by the time elapsed from the beginning to final position; (2) straightness ratio (directionality ratio) - calculated by the straight-line distance between the start and end point of the cell track, divided by the track length. A straight cell trajectory equates to a straightness ratio of 1, a curved trajectory equates to a ratio of 0; (3) mean squared displacement (MSD) - based on particle movement theory, can be used to characterise the type of motion displayed by cells[87] and measures the average distance travelled by a cell over a specified time interval. Nikon NIS-Elements calculates MSD as a sum of the squared distances from the beginning of the track.

**KRT5$^+$ cell migration through CDMs.** Images captured using the Cy5 and brightfield channels of the ZEISS Axio Observer were pre-processed in Icy[86] (https://icy.bioimageanalysis.org/). Z-stack maximum intensity projection was performed on the 12-h image sequence (144 frames, 5- min intervals). In Nikon NIS Elements v.4.50 the image sequence was calibrated (xy pixel size of 1.30 µm, and time step of 300 s) and images aligned to previous frame to correct for drift. Spot detection was performed on the Cy5 channel with the detector set to detect bright, circular objects of different sizes of typical diameter 27 µm with contrast set to 140. Dark objects were removed. Automated tracking was optimised using a random motion, constant speed model. A maximum gap size of 3 was allowed between tracks. Tracks were terminated when no object was closer than 2 times the standard deviation from a predicted position. A maximum object speed of 100 µm/s was set to exclude rapidly moving floating debris. Tracks with 12 or less frames were deleted. The automated tracks were manually checked for accuracy. The raw cell migration data generated by Nikon NIS-Elements software was exported to MotilityLab (http://www.motilitylab.net/) and GraphPad Prism v9.4.1 for further analysis. Only cells tracked for the entire 12-h duration were included in the final analysis.

**KRT5$^+$ cell migration on ECM proteins identified through mass spectrometry-proteomics.** As previously described, images captured using the brightfield channels of the ZEISS Axio Observer were pre-processed in Icy[86] (https://icy.bioimageanalysis.org/). Z-stack maximum intensity projection was performed on the 12-h image sequence (144 frames, 5- min intervals). In Nikon NIS- Elements the image

sequence was calibrated (xy pixel size of 1.30 μm, and time step of 300 s) and images aligned to previous frame to correct for drift. Spot detection was performed on the brightfield channel with the detector set to detect dark spots, 7 px. Other settings were as per the CDM experiments. The raw cell migration data generated by Nikon NIS-Elements software was exported to MotilityLab (http://www.motilitylab.net/) and GraphPad Prism v9.4.1 for further analysis and presentation. Only cells tracked for the entire 12-h duration were included in the final analysis.

## Immunofluorescence microscopy of KRT5⁺ cells

**Submerged culture.** KRT5⁺ cells in media (PromoCell, supplemented as previously) were seeded (3.0 ×10⁴/ well) on a collagen I (PureCol, Advanced BioMatrix, #5005) coated 96-well μ-plate (Ibidi). When confluent, the cells were fixed in 4% PFA (Electron Microscopy Sciences, #RT-15710) for 20 min, washed in PBS, then permeabilised with permeabilisation buffer (eBioscience; #00-8333-56) for 20 min (room temperature). The cells were blocked with 2% BSA in PBS for 30 min (room temperature) then incubated with primary conjugated antibodies to KRT5 overnight at 4 °C. Cell nuclei were counterstained with DAPI (Sigma-Aldrich #D9542; 1:2000). Imaging was carried out with a Leica SP5 inverted confocal microscope. Images were acquired using LAS AF (Leica) and analysed using Fiji v1.52p software[81].

**High-resolution Airyscan microscopy.** UV-sterilised glass coverslips were pre-coated in ECM proteins as previously described. KRT5⁺ cells were seeded at a density of 1.0 ×10⁴ cells per well. After 24 h of incubation these cells were fixed in 4% formaldehyde (Electron Microscopy Sciences). Phalloidin Alexa Fluor 488 (Invitrogen) was used to stain actin cytoskeleton and DAPI (Sigma-Aldrich #D9542; 1:1000) as a nuclear counterstain. The cells were coverslip mounted with ProLong Gold Antifade Mountant (Invitrogen; #P36934). High-resolution images were acquired using a 40 × 1.3 NA oil objective on ZEISS 880 microscope using Airyscan detector (Beatson Institute, Glasgow, UK). Airyscan image processing was carried out using default settings in ZEISS ZEN v2.3 (Black Edition).

## Wound healing PCR array

KRT5⁺ cells were lysed by the addition of 350 μl Buffer RLT (Qiagen, #79216) containing 1% β-mercaptoethanol (β-ME). RNA was extracted using the RNeasy MiniKit (Qiagen) as per manufacturer's instructions. RNA concentration and RNA integrity number (RIN) was confirmed using 1 μl of the RNA sample with the Agilent RNA ScreenTape assay and Agilent 2200 TapeStation instrument and software, as per manufacturer's protocol. The expression of a panel of genes involved in wound healing pathways, was evaluated by real-time RT-PCR using an RT² Profiler PCR Array (Qiagen, #PAHS-121Z) in a 96-well plate format. Each plate contained primer assays for 84 wound healing genes and 5 housekeeping genes for normalisation of the data. They also incorporated controls to test for genomic DNA contamination, and the efficiency of the reverse-transcription and PCR reactions. A total of 300 ng RNA from each sample was used to synthesise cDNA with the RT² First Strand Kit (Qiagen, # 330401), incubating at 42 °C for 15 min, then at 95 °C for 5 min in a Veriti 96-well Thermal Cycler (Applied Biosciences). The cDNA was mixed with the SYBR Green ROX qPCR Mastermix (Qiagen) and aliquoted into the wells of the RT² Profiler PCR Array. PCR was performed on a ViiA 7 instrument (Applied Biosystems) using Fast 96-well block. The following cycling conditions were programmed; 1 cycle at 95 °C for 10 min; 40 cycles at 95 °C for 15 s and 60 °C for 1 min, with a ramp rate adjusted to 1 °C/s. The real-time PCR threshold cycle (C_T) values were analysed using the Qiagen GeneGlobe Data Analysis Center (https://geneglobe.qiagen.com/it/analyze). Within this software, sample quality control was performed, and the housekeeping gene(s) displaying smallest variance across samples was selected. A C_T cut-off of 35 was set for the analysis. Normalised gene expression values are calculated as the difference between the $C_T$ value of the gene of interest (GOI) and the average $C_T$ value of the selected housekeeping genes (HKG):

$$2^{-[C_T(GOI)-CT(AVG\ HKG)]} = 2^{-\Delta C_T}$$

Fold- change is calculated using the delta delta CT method[88] by dividing the normalised expression in IPF samples by the normalised expression in healthy control samples:

$$\frac{2^{-\Delta C_{T(IPF)}}}{2^{-\Delta C_{T(CON)}}} = 2^{-\Delta\Delta C_T}$$

A Student's $t$ test of the replicate normalised gene expression values for each gene in the IPF group and healthy control groups was performed. PCA and hierarchical clustering was performed in ClustVis[89] (https://biit.cs.ut.ee/clustvis/), and volcano plots were generated using VolcaNoseR[90] (https://huygens.science.uva.nl/VolcaNoseR/).

## Mass spectrometry (MS)- based proteomics

**Isolation of enriched cell-derived ECM and MS sample preparation.** A method developed by Lennon et al. [37] was used to reduce the complexity of protein samples for MS analysis by removing cellular components and enriching for ECM proteins. All steps were carried out at 4 °C to minimise proteolysis. Primary HLFs from 6 donors ($n = 3$ controls, $n = 3$ IPF) were cultured in a 10cm² petri dish and maintained for 21 days in DMEM (Gibco) and 10% FBS to allow matrix deposition. Following a PBS wash, the cells were incubated for 30 min in extraction buffer (10 mM Tris, 150 mM NaCl, 1% [vol/vol] Triton X-100, 25mMEDTA, protease inhibitor) to solubilise cellular proteins, and samples were then centrifuged at 14,000xg for 10 min. The remaining pellet was incubated for 30 min in alkaline detergent buffer (20 mM NH4OH and 0.5% [vol/vol] Triton X-100 in PBS) to further solubilise cellular proteins and disrupt cell−ECM interactions. Samples were then centrifuged at 14,000 × $g$ for 10 min. The remaining pellet was resuspended in S trap lysis buffer (5% SDS with 50 mM TEAB pH 7.5) and transferred to a 130 μl AFA tube (Covaris Inc, Woburn, USA,Covaris) to yield the ECM fraction. The lysate was further solubilised using AFA sonication in a Covaris LE220+ (Woburn, Massachusetts). Sample were disrupted for 300 s, with 50 cycles per burst, a duty factor of 20% and a peak incident power of 500 W.

**S-Trap™ proteolytic digestion.** The resulting ECM-enriched fractions were reduced in 5 mM dithiothreitol (DTT) and incubated for 10 min at 60 °C to reduce the cysteine bonds. The samples were then alkylated in 15 mM iodoacetamide (IAM) for 30 min (in the dark). The alkylation reaction was finally quenched using 5 mM DTT. Aqueous phosphoric acid was added to the SDS lysates to a final concentration of 1.2% then S-Trap binding buffer (90% MeOH, 100 mM final TEAB, pH 7.1) was mixed with the acidified protein lysates. The acidified lysate/S-Trap buffer mix was loaded onto the S-Trap column. The captured proteins were washed centrifuging through 150 μl of S-Trap binding buffer at 4000 × $g$ for 2 min and repeated three times. The proteins were digested by adding sequencing grade modified trypsin (Promega, #V5113) at a ratio of 1:10 wt:wt (enzyme:protein) to the top of the protein trap and incubated in an Eppendorf Thermomixer for 1 hr at 47 °C. The peptides were eluted with 65 μl of digestion buffer (50 mM TEAB) from the S-Trap column and centrifuged at 4000 × $g$ for 2 min then 65 μl of 0.1% aqueous formic acid (FA) was added and centrifuged at 4000 × $g$ for 2 min. The remaining hydrophobic peptides were eluted with 30 μl of 30% aqueous acetonitrile containing 0.1% formic acid and centrifuged at 4000 × $g$ for 2 min. All three elutions were combined and peptides were desalted using 1 mg of R3 beads which were placed in each well of a 96 well plate with 0.2 um PVDF

membrane. R3 beads were washed with 50% (v/v) acetonitrile followed by 0.1% (v/v) formic acid. Peptide samples were then resuspended in 5% (v/v) acetonitrile and 0.1% (v/v) formic acid to allow peptide binding to beads, washed twice with 0.1% (v/v) formic acid, and peptides eluted in 50% (v/v) acetonitrile, 0.1% (v/v) formic acid. Peptides were desiccated in a vacuum centrifuge and resuspended in 10 µl of 5% (v/v) acetonitrile and 0.1% (v/v) formic acid.

**MS data acquisition.** Digested samples were analysed by LC-MS/MS using an UltiMate® 3000 Rapid Separation LC (RSLC, Dionex Corporation, Sunnyvale, CA) coupled to a QE HF (Thermo Fisher Scientific, Waltham, MA) mass spectrometer. Mobile phase A was 0.1% formic acid in water and mobile phase B was 0.1% formic acid in acetonitrile and the column used was a 75 mm × 250 µm i.d. 1.7 µM CSH C18, analytical column (Waters). A 1 µl aliquot of the sample was transferred to a 5 µl loop and loaded onto the column at a flow of 300 nl/min for 5 min at 5% B. The loop was then taken out of line and the peptides separated using a gradient that went from 5% to 7% B and from 300 nl/min to 200 nl/min in 1 min followed by a shallow gradient from 7% to 18% B in 64 min, then from 18% to 27% B in 8 min and finally from 27% B to 60% B in 1 min. The column was washed at 60% B for 3 min before re-equilibration to 5% B in 1 min. At 85 min the flow is increased to 300 nl/min until the end of the run at 90 min. Mass spectrometry data was acquired in a data directed manner for 90 min in positive mode. Peptides were selected for fragmentation automatically by data dependant analysis on a basis of the top 12 peptides with m/z between 300 to 1750Th and a charge state of 2, 3 or 4 with a dynamic exclusion set at 15 s. The MS Resolution was set at 120,000 with an AGC target of 3e6 and a maximum fill time set at 20 ms. The MS2 Resolution was set to 30,000, with an AGC target of 2e5, a maximum fill time of 45 ms, isolation window of 1.3Th and a collision energy of 28.

**MS data analysis.** Raw MS files were imported into Proteome Discoverer v2.4.1.15 SP1 (PD) (Thermo Scientific), a program that integrates all the different steps in a quantitative proteomics experiment (including MS/MS spectrum extraction, peptide identification and quantification) into automated workflows. The data were analysed using the standard Sequest™ HT-Percolator workflow and the default consensus workflow for label-free quantification. The Sequest HT search was performed against the Homo sapiens database (SwissProt and TrEMBL version 2017-10-25, taxonomy 9606, 42252 and 124124 sequences respectively).

The hierarchical clustering of the proteomic data and the principal component analysis plots were computed and visualised using ClustVis[89] (https://biit.cs.ut.ee/clustvis/). Proteome Discoverer v2.4.1.15 SP1 (PD) (Thermo Scientific) uses the ANOVA method to calculate the statistical significance of differential protein expression between sample groups, with a lower P value indicating the abundances are likely to be statistically different. The Benjamini-Hochberg procedure was used to adjust for multiple comparisons. Protein interaction network analysis was performed using STRING[41] v11 (https://string-db.org/), a database of known and predicted protein-protein interactions. The list of matrix proteins identified, and their respective fold change (based on calculated abundance ratios) were submitted to the search under proteins with values/ranks. The proteins were mapped onto the Homo sapiens network, with a FDR stringency of 1%. The active interaction sources selected were experiments and co-expression databases and the network edges represents the confidence (strength of data support). A medium confidence score of 0.4 was applied. STRING uses a scoring system to indicate the confidence level for each protein-protein interaction. More information on how this is calculated is provided in ref. 41.

The sample preparation and mass spectrometry analysis was supported by the Biological Mass Spectrometry (BioMS) facility in the Faculty of Biology, Medicine and Health of the University of Manchester.

## Statistics and reproducibility

Statistical analysis was carried out using GraphPad Prism v9.4.1 (GraphPad Software, LLC.). To determine if the data was from a Gaussian distribution the D'Agostino-Pearson omnibus normality test was performed. If the p-value returned for this test was high (>0.05), it was assumed the data were sampled from a Gaussian distribution and a parametric test was used for statistical analysis; for unpaired data this was an unpaired t test, and for paired data this was a paired t-test. If the normality test p-value was ≤0.05, the null hypothesis was rejected and therefore data were not sampled from a Gaussian distribution so a nonparametric test was used; for unpaired data this was a Mann–Whitney test and for paired data this was a Wilcoxon matched-pairs signed rank test. For cell migration experiments where multiple cells from multiple donors were studied under different experimental conditions, a nested one-way ANOVA with Tukey's multiple comparison test was used. For quantitative analysis of SHG data acquired from different lung regions from multiple IPF and control donors, an ordinary one-way ANOVA with Tukey's multiple comparisons test was performed. For correlation analyses, Spearman correlation coefficients were computed and a two-tailed P value, ≤0.05 considered significant. The statistical test used for each analysis are described in the accompanying figure legend and significant differences noted by asterisk(s): *P < 0.05, **P < 0.01, ***P < 0.001, ****P < 0.0001. The number of biological replicates are indicated in each figure legend. Data was replicated across the samples tested. For image texture analysis (Fig. 1) each data point represents the average of 2–6 images per area per control (n = 5) and IPF patient (n = 8 for PB, PVS, ALV, FIBRO and n = 6 for HC). For the cell migration experiments the total number of individual cells tracked per group are indicated in the results section. For the CDM cell migration work, 2 technical replicates per donor were used.

## Reporting summary

Further information on research design is available in the Nature Portfolio Reporting Summary linked to this article.

## Data availability

The mass spectrometry proteomics data generated in this study have been deposited to the ProteomeXchange Consortium via the PRIDE[91] partner repository with the dataset identifier PXD037236. Raw image files are stored on servers at Imperial College London due to their large file size and are available from the corresponding author on request. Source data are provided with this paper.

## Code availability

The scripts used for the image analyses can be found at https://doi.org/10.5281/zenodo.8288989. Principal component analysis (PCA) plots were generated in RStudio using freely available code (CC BY-NC-SA 3.0 US): http://www.sthda.com/english/articles/31-principal-component-methods-in-r-practical-guide/112-pca-principal-component-analysis-essentials/. The following Fiji/ ImageJ pluggins were used for image texture analysis: Texture Analyzer (https://imagej.nih.gov/ij/plugins/texture.html) to perform GLCM analysis and OrientationJ (http://bigwww.epfl.ch/demo/orientation/#measure) to measure the orientation of collagen fibres.

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

## Acknowledgements

This work was supported by the Imperial College Faculty of Medicine Clinician-Investigator Scholarship Award and NHLI Clinical Lecturer award (R.J.H.); NIHR Imperial Biomedical Research Centre (BRC) Respiratory Shadow Theme project award P89068 (F.P.); Action for Pulmonary Fibrosis Mike Bray Fellowship (P.L.M.); Wellcome Senior Fellowship award 107059/Z/15/Z (C.M.L.); NIHR Clinician Scientist Fellowship CS-2013-13-017 and British Lung Foundation Chair in Respiratory Research C17-3 (T.M.M.); Wellcome Senior Fellowship award 202860/Z/16/Z (R.L.); Kidney Research UK grant RP_040_20180306 (M.F.); Wellcome Trust 203128/Z/16/Z (to the Wellcome Centre for Cell-Matrix Research at the University of Manchester); Cancer Research UK (CRUK) to the CRUK Beatson Institute A31287 (F.F. and L.M.C.) and core funding from CRUK A23983 (L.M.C.);. The Facility for Imaging by Light Microscopy (FILM) at Imperial College London is part-supported by funding from the Wellcome Trust grant 104931/Z/14/Z and BBSRC grant BB/L015129/1. We are grateful to the Beatson Advanced Imaging Resource for microscopy support. We thank clinical colleagues from the Royal Brompton Hospital Interstitial Lung Disease (ILD) Unit for supporting this work. We thank staff from Biological Mass Spectrometry Core Facility at the University of Manchester for their assistance. We thank Lorraine Lawrence of the Research Histology Facility (NHLI, Imperial College London) for her assistance. We thank the patients and healthy volunteers who generously donated samples for this study. We also acknowledge the National Institute for Health Research (NIHR) Biomedical Research Centre based at Imperial College Healthcare NHS Trust and Imperial College London. The views expressed are those of the author(s) and not necessarily those of the NHS, the NIHR or the Department of Health.

## Author contributions

Conceptualisation, R.J.H., A.J.B., T.M.M. and C.M.L.; Methodology, R.J.H., F.P., R.L., L.M.C., A.J.B. and C.M.L.; Formal Analysis, R.J.H., F.P., D.C.A.G., F.F., M.F., E.R.; Investigation, R.J.H., F.P., D.C.A.G., F.F., M.F., W.T., L.Y., S.A.W.; Resources, R.J.H., L.Y., S.A.W., P.L.M., S.V.K., A.N., A.R., and T.M.M; Writing – Original Draft, R.J.H., C.M.L., Writing – Review & Editing, R.J.H., P.L.M., R.L., E.R., L.M.C., A.J.B., T.M.M. and C.M.L.; Funding Acquisition, R.J.H., C.M.L.; Supervision, A.J.B., T.M.M. and C.M.L.

## Competing interests

R.J.H. received, unrelated to the submitted work, consultancy fees from Boehringer Ingelheim. P.L.M. received, unrelated to the submitted work, industry-academic funding from AstraZeneca via his institution and speaker and consultancy fees from Boehringer Ingelheim Trevi and Hoffman-La Roche. A.J.B. received, unrelated to the submitted work, consultancy fees and/or industry/academic funding from Ammax, Devpro, and Ionis pharmaceuticals, via his institution. A.G.N. is or has been a scientific advisor relating to IPF trials for Medical Quantitative Image Analysis, Galapagos, Boehringer Ingelheim and Roche, as well as receiving payment for educational activities relating to interstitial lung disease from Boehringer Ingelheim and UpToDate. T.M.M. received, unrelated to the submitted work, consultancy fees in relation to pulmonary fibrosis from Abbvie, Agomab, Apellis, Astra Zeneca, Bayer, Biogen Idec, Blade Therapeutics, BMS, Boehringer Ingelheim, Bridge Therapeutics, Carthronix, Chiesi, CohBar, CSLBehring, Daewoong, Daiatchi, DevPro, Endeavor, Fibrogen, Galapagos, Galecto, GlaxoSmithKline, Insilico, IQVIA, Kinevent, Pliant, Pfizer, Puretech, Qureight, Redx, Remedy Cell, Respivant Sciences, Roche, Shinogi, Surrozen, Theravance, Three Lakes Partners, Trevi, UCB, United Therapeutics, Veracyte, Vicore. The remaining authors declare no competing interests.
