## [Peer Review File · Nature Communications]

Lung extracellular matrix modulates KRT5+ basal cell activity in pulmonary fibrosisREVIEWER COMMENTS

Reviewer #1 (Remarks to the Author):

Idiopathic pulmonary fibrosis (IPF) is a chronic progressive disease that eventually leads to respiratory failure. IPF is thought to be mediated by repeated lung injury, leading to irreversible damage to the millions of alveoli, which are the tiny distensible air sacs at the end of the bronchioles. Unfortunately, the molecular and cellular mechanisms that result in the progressive loss of tissue homeostasis associated with IPF disease are not well characterized. The authors and others previously demonstrated that multipotent KRT5+ BCs, which normally reside in the conducting airways, accumulate within the fibrotic and honeycomb (HC) regions associated with the respiratory airway zones of lung tissue derived from IPF patients. It is interesting that the authors report a topographical difference in collagen fibers between fibrotic and HC regions, the former being more disorganized, thus they likely have a differential impact on cellular signaling and behaviors in these microenvironments. Data presented in Figure 2 on the close proximity of the KRT5+ BCs and fibroblasts are supportive of their interpretation of potential fibroblast- KRT5+ BCs crosstalk that may promote IPF progression within the fibrotic Collagens I and III, and proteoglycan rich- microenvironments. Data in Figure 3 demonstrate that KRT5+ BCs exhibit migratory behavior on fibrillar collagen matrices, which likely serve as the primary migratory substratum in vivo for their translocation from the conducting airways to fibrotic and HC regions. Collagen and Fibronectin (FN) are two major fibrillar components of ECMs that promote cell adhesion and migration. An abundance of FN was demonstrated in the purified CDMs (Cell-Derived Matrices) derived from control and IPF fibroblasts (Figure 4). I agree with the authors' statement that these two CDMs are likely to mimic some of the ECM complexity of the healthy and fibrotic 3D matrix microenvironments in vivo. Hence, the data generated is likely reflective of cellular events occurring in vivo. They also present evidence that the secretome of IPF-derived fibroblasts has a distinct molecular signature from that generated by fibroblasts isolated from healthy individuals. Importantly, the authors convincingly demonstrate that SPARC, a collagen-binding multifunctional glycoprotein, is one of a select group of proteins abundant in IPF CDMs. Importantly, SPARC was the only one to restrict the migratory behavior of healthy KRT5+ BCs when seeded on a composite Collagen I/SPARC matrix relative to control Collagen I. While it is hypothesized that SPARC may serve to promote the retention of KRT5+ BCs within the fibrotic microenvironments, SPARC is a counter-adhesive glycoprotein. Thus, SPARC may facilitate durotaxis-like mediated migration of KRT5+ BCs from distal sites to the dense fibrotic areas of disease progression.

The results presented in the manuscript are novel and follow a logical progression with conclusions based on solid experimental evidence. The quality of the multiplexed fluorescent images is particularly noteworthy. In summary, this is a well-written and interesting manuscript that contributes to our basic understanding of the cellular and matrix dynamics associated with the fibrotic and HC regions of IPF patients. This information is likely to make an important contribution in the development of therapeutics for the treatment of IPF and potentially other fibrotic diseases.

Additional comments:

It is likely that KRT5+ BCs that accumulate in fibrotic and HC regions of IPF lungs are derived from the airway epithelium. However, human alveolar type 2 epithelial cells have been reported to be capable of transdifferentiating into KRT5+ BCs during alveolar repair. Hence, the authors should comment in the discussion on whether it is possible KRT5+ BCs are derived from two sources.

There is no indication of the commercial sources of IGFBP-5, GAS6, FN1, and SPARC. While the data presented in Figure 7 is consistent with the source of SPARC associating with the Collagen I substratum, two SPARC glycoforms exist in vertebrates, only one of which has an affinity for collagens. Loss of N-glycosylation increases the affinity of SPARC for fibrillar collagens. Is the SPARC used as a recombinant that is glycosylated or unglycosylated? In the extended data Figure 6, different

concentrations of IGFBP-5, GAS6, and FN1 were assayed. Were different concentrations of SPARC also tested, which accounts for why a SPARC concentration of 20 ug/ml was selected for the experiments?

Reviewer #2 (Remarks to the Author):

The manuscript by Hewitt et al. provides a variety of state of the art techniques used to evaluate differences in cell behavior in normal versus IPF conditions. Specifically, activities associated with KRT5+ basal cells was addressed in terms of responses to changes in extracellular matrix. As fibrosis is a hallmark of IPF, this is a relevant and important aspect to address. The images and the quality of the movies were exceptional and add nicely to the paper. The technological advances used to address ECM composition, organization, and cell response to ECM are laudable. While the results are interesting and contribute to the field, there are a number of critically important aspects to address prior to publication.

- 1) Fig. 1: Was any attempt made to localize SPARC in tissue sections from IPF lungs? One would perhaps anticipate that higher levels of SPARC would be found in regions where KRT5+ cells associate with fibrosis, i.e. honeycomb and peribronchial areas. There are a number of anti - SPARC antibodies available that work well in human tissue sections (e.g., Sweetwyne et al, JHC 2016).
- 2) Fig 2: Was any attempt made to localize fibronectin in IPF vs. normal lungs as fibronectin appears to provide the best substrate for KRT5+ cell migration in CDM (Fig. 5)?
- 3) Fig. 4: The most critical weakness in these studies is the lack of ascorbate in the generation of CDM. The methods state that ascorbate was not added to cultures because ascorbate would promote cross-linking of collagen and that this would artificially alter the properties of the matrix in a way that would influence cell migration or MS-proteomic results. First, because all fibrillar collagen is cross-linked in vivo, as this is a requirement for incorporation into an insoluble ECM, this reviewer would argue that a cross-linked ECM is a more valid construction to test cell response to ECM. But, more importantly, ascorbic acid is not primarily a mediator of collagen cross-linking. The mechanism of ascorbic acid is to serve as a co-factor for prolyl hydroxylase. Hydroxylation of prolines is required for efficient post-translational modification of not only fibrillar collagens, but basement membrane collagen IV as well. In the absence of sufficient ascorbic acid, collagens can be retained in the ER and Golgi and subject to increased intracellular degradation (this is true for all collagen producing cells and was specifically addressed in human lung fibroblasts by Berg et al. Arch. Biochem. Biophys. 1983). Less collagen will be secreted and less properly modified collagens will be available for proper incorporation into the ECM. This will make for a very different ECM composition than what is produced in vivo. The lack of ascorbic acid provided to cells generating CDM is considered to be a significant flaw and hampers the interpretation of the experimental results of these studies.
- 4) Fig. 5, it was difficult to reconcile the differences in panel e, where no significant differences were found in cell displacement, versus panel h. Please provide a better explanation as to how mean square displacement differs from cell displacement and why this would generate divergent results.
- 5) Fig. 6: panel g – SPARC is well-characterized as a counter-adhesive protein (e.g. Sage et al, J. Cell Biol. 1989). It is not accurate to place SPARC in the Venn diagram between cell migration and ECM organization with exclusion from cell adhesion.
- 6) Fig. 7: what is the rationalization for using only KRT5+ cells from normal lung in cell migration assays in Fig. 7? Wouldn't IPF KRT5+ cells be the more relevant cell type to use? Or a comparison between the normal and IPF perhaps showing equivalent (or not equivalent) effects in normal versus IPF cells?
- 7) SPARC is a collagen binding protein. That addition of SPARC to cells plated and migrating on collagen would impede collagen-dependent migration is not particularly surprising. That the Discussion states that, "Our data are the first to show a function for SPARC in human tissue" is over-stated. First, a function of SPARC in human tissue was shown for by Mendoza-Londono et al (Am J Human Gen. 2015) where missense mutations in SPARC were shown to be causative in osteogenesis imperfecta. Second, showing that SPARC inhibits collagen-dependent migration in an in vitro assay is not

conclusive of a function of SPARC in human tissue. Perhaps, recommend modifying the statement to reflect that the data strongly support a functional activity for SPARC in lung/soft, connective tissue, for the first time.

Minor points:

- a) The source of recombinant SPARC is not given in the list of reagents.
- b) The biorender generated diagrammatic panels do not lend a great deal to the figures.
- c) Fig. 7, panel G – the quality of this summary figure could be improved. For example, at one point in the manuscript, the authors mention cell:cell close proximity and perhaps contact with KRT5+ cells and fibroblasts. As should be indicated, the ECM is a complex composition of proteins, why would the summary figure depict a grossly simple structure with the word SPARC interspersed in the gray lines when the entire paper is devoted to a better understanding of the complexities of the ECM?

Reviewer #3 (Remarks to the Author):

In their study “Lung extracellular matrix modulates KRT5+ basal cell activity in pulmonary fibrosis” by Hewitt et al., the authors investigate the migratory behaviour of KRT5+ (basal) cells (BC) in healthy lung tissue and that of patients with idiopathic pulmonary fibrosis (IPF). To this end, they use a variety of methods to assess the structure and composition of the extracellular matrix including 2D and 3D migration assays, second harmonic generation 2-photon microscopy, imaging mass cytometry and mass cytometry. In their study, they find differences in collagen deposition and structure between the different disease conditions and observe different migratory behaviours between KRT5+ BC and KRT5+ cells found in IPF. They then also compared the gene expression profiles of healthy and IPF KRT5+ cells when grown on either collagen I coated 2D assays or 3D cell-derived matrix and observed major differences. Notably, collagens were upregulated in all KRT5+ cells cultured on IPF derived matrix. Assessment of matrix derived from healthy and IPF tissues then showed the upregulation of 14 matrix proteins associated with matrix remodelling. The effect of five of these 14 upregulated matrix proteins on KRT5+ cell migration was then assessed in 2D migration assays. This revealed the inhibitory effect of SPARC1 on the migration of KRT5+ basal cells. The authors show that different ECM compositions appear to have an effect on the migratory behaviour of KRT5+ cells, however there are several concerns regarding this study, especially regarding the cogency of experiments from which the conclusion is being drawn that SPARC is the key player in KRT5+ cell migration in IPF.

Major concerns:

1. Generally, formal statistical comparisons are missing, often between IPF and healthy control tissue. This especially concerns figures: 1c,3i In lines 126-127 it is stated that “higher collagen density was evident “. However, while other significant p-values are marked on the figure, this comparison was either not included or was not significant.
2. In figure 1d, it is not clear from the text which different conditions or tissue sites are being referred to. In lines 127-129 it is stated that “IPF HC regions clustered with the peribronchial regions of IPF and controls due to increased collagen density (Fig. 1e) and coherency of fibre orientation (Fig. 1f) within these areas.”. IPF HC regions are not labelled on the corresponding PCA plot in Figure 1d.
3. In figure 1h, Pearson correlation is being used to correlate the number of KRT5+ cells /mm² with collagen density. ...
4. Imaging mass cytometry was used to acquire multiplexed images of one healthy control sample and one IPF sample. KRT5+ is being used to identify KRT5+ (basal) cells. Yet, a nuclear marker is missing to properly identify cells within the measure area. Given the highly multiplexed nature of the method, it would have been desirable to include more markers for both the ECM as well as other cell types in the tissue, including fibroblasts, which were identified to be vital for KRT5+ cell migration by the authors. Despite their efforts to visualise SMA+ fibroblasts and KRT5+ cells using high-resolution microscopy, this analysis could be done more efficiently by studying the neighbouring cells of KRT5+ cells between healthy and diseased tissues, by performing a neighbourhood analysis for example.

Otherwise, the statement in lines 165-166 "Signalling between these neighbouring cells may drive pathology in the IPF lung." remains highly speculative. Many more ECM proteins are identified as important throughout this study, including FN1 for example. Thus, it would be interesting to see these markers included for ICM imaging. It would also be interesting to analyse whether the described effect of ECM protein distribution patterns is valid across the entire cohort of patients.

5. Despite it being mentioned several times later on in the manuscript, it is unclear why FN1 was not included in the different conditions for the migration assay.

6. It is unclear to the reader whether the migration assays were done repeatedly and how different many patient samples were involved.

7. For the 2D migration assays, all 3 collagens and Versican were tested individually. Seeing how the ECM is a mixture of a multitude of matrix proteins, it would be interesting to test the individual behaviours against a mixture of all four components, perhaps even with varying concentrations. (Figure 3e). In figures 3f-h, a formal comparison between IPF and healthy samples would be desirable.

8. Since the KRT5+ cells show different migration patterns in 2D and 3D, it would be interesting to know whether the KRT5+ cells also migrated through the matrix in the 3D assay.

9. Seeing how KRT5+ cells also contribute to the ECM production (Figure 4k), it is unclear to the reader, if the described effects of changes in the ECM composition are related to the KRT5+ cells themselves or really the surrounding fibroblasts.

10. It is unclear why 2D assays were chosen in figure 7 to assess the impact of SPARC1 on KRT5+ cell migration and why this experiment was not repeated with IPF derived KRT5+ cells.

11. It is hypothesised that SPARC is being produced by human lung fibroblasts. However, functional assessment is missing to support this hypothesis. Since the authors use fibroblast derived matrix in their experiments, this could be further analysed using SPARC knock-out models.

Minor concerns:

1. Instead of only showing healthy control lung samples in the supplementary figure, it would be beneficial for direct comparison to show both healthy and diseased tissue in figure 1. It would further also be helpful to the reader to label the healthy and diseased tissue images as such in the figures.

2. Figure legends are often only displayed once in a figure with multiple panels but should be added to each figure panel for easier and faster readability.

3. The genes in figures 4j and 6e are not readable.

4. It is also unclear what each cluster is formed of in figure 6e. For better readability, it would be easier to include a .csv file with all differentially expressed genes per cluster and to highlight only the most important ones per cluster. There are also considerable differences between the clusters which are not explained. In figures 6e-f, a legend is missing explaining the different colours. The heatmap counts bar is also missing a label.

5. Figure 6d is barely readable.

6. Figure 7e is missing a legend and axis. All other migration assay plots are also mixing axis descriptions.

7. Capitalisation is inconsistent throughout all figures (e.g., log₂ (Figure 6b)– Log₂ (Figure 6f))

REVIEWER COMMENTS

Reviewer #1 (Remarks to the Author):

Idiopathic pulmonary fibrosis (IPF) is a chronic progressive disease that eventually leads to respiratory failure. IPF is thought to be mediated by repeated lung injury, leading to irreversible damage to the millions of alveoli, which are the tiny distensible air sacs at the end of the bronchioles. Unfortunately, the molecular and cellular mechanisms that result in the progressive loss of tissue homeostasis associated with IPF disease are not well characterized. The authors and others previously demonstrated that multipotent KRT5+ BCs, which normally reside in the conducting airways, accumulate within the fibrotic and honeycomb (HC) regions associated with the respiratory airway zones of lung tissue derived from IPF patients. It is interesting that the authors report a topographical difference in collagen fibers between fibrotic and HC regions, the former being more disorganized, thus they likely have a differential impact on cellular signaling and behaviors in these microenvironments. Data presented in Figure 2 on the close proximity of the KRT5+ BCs and fibroblasts are supportive of their interpretation of potential fibroblast- KRT5+ BCs crosstalk that may promote IPF progression within the fibrotic Collagens I and III, and proteoglycan rich-microenvironments. Data in Figure 3 demonstrate that KRT5+ BCs exhibit migratory behavior on fibrillar collagen matrices, which likely serve as the primary migratory substratum in vivo for their translocation from the conducting airways to fibrotic and HC regions. Collagen and Fibronectin (FN) are two major fibrillar components of ECMs that promote cell adhesion and migration. An abundance of FN was demonstrated in the purified CDMs (Cell-Derived Matrices) derived from control and IPF fibroblasts (Figure 4). I agree with the authors' statement that these two CDMs are likely to mimic some of the ECM complexity of the healthy and fibrotic 3D matrix microenvironments in vivo. Hence, the data generated is likely reflective of cellular events occurring in vivo. They also present evidence that the secretome of IPF-derived fibroblasts has a distinct molecular signature from that generated by fibroblasts isolated from healthy individuals. Importantly, the authors convincingly demonstrate that SPARC, a collagen-binding multifunctional glycoprotein, is one of a select group of proteins abundant in IPF CDMs. Importantly, SPARC was the only one to restrict the migratory behavior of healthy KRT5+ BCs when seeded on a composite Collagen I/SPARC matrix relative to control Collagen I. While it is hypothesized that SPARC may serve to promote the retention of KRT5+ BCs within the fibrotic microenvironments, SPARC is a counter-adhesive glycoprotein. Thus, SPARC may facilitate durotaxis-like mediated migration of KRT5+ BCs from distal sites to the dense fibrotic areas of disease progression.

The results presented in the manuscript are novel and follow a logical progression with conclusions based on solid experimental evidence. The quality of the multiplexed fluorescent images is particularly noteworthy. In summary, this is a well-written and interesting manuscript that contributes to our basic understanding of the cellular and matrix dynamics associated with the fibrotic and HC regions of IPF patients. This information is likely to make an important contribution in the development of therapeutics for the treatment of IPF and potentially other fibrotic diseases.

Additional comments:

It is likely that KRT5+ BCs that accumulate in fibrotic and HC regions of IPF lungs are derived from the airway epithelium. However, human alveolar type 2 epithelial cells have been reported to be capable of transdifferentiating into KRT5+ BCs during alveolar repair. Hence, the authors should comment in the discussion on whether it is possible KRT5+ BCs are derived from two sources.

Thank you for this comment. We agree and have updated the discussion accordingly with the following text:

“The origin of KRT5+ BCs that accumulate in the distal IPF lung tissue is debated. Murine lineage-tracing studies have shown that following alveolar injury, KRT5+ cells arise from progenitor populations in the airway to repopulate damaged areas of lung^{12, 65, 66}. Given species- related differences in cellular composition of the distal airways, it is unclear how murine studies translate to human disease⁶⁷. Single-cell transcriptional profiling revealed hypoxia and activated Notch signalling as drivers of a KRT5+ basal-like phenotype in human AT2 cells from fibrotic lungs⁶⁸. Organoid and co-culture models have demonstrated the capacity of human (but not murine) AT2 cells to transdifferentiate into KRT5+ cells with cues from mesenchymal cells¹¹. It is possible that KRT5+ BCs in the distal fibrotic lung are derived from more than one source.” (Lines 488 – 497, Page 13, marked copy)

There is no indication of the commercial sources of IGFBP-5, GAS6, FN1, and SPARC. While the data presented in Figure 7 is consistent with the source of SPARC associating with the Collagen I substratum, two SPARC glycoforms exist in vertebrates, only one of which has an affinity for collagens. Loss of N-glycosylation increases the affinity of SPARC for fibrillar collagens. Is the SPARC used as a recombinant that is glycosylated or unglycosylated? In the extended data Figure 6, different concentrations of IGFBP-5, GAS6, and FN1 were assayed. Were different concentrations of SPARC also tested, which accounts for why a SPARC concentration of 20 ug/ml was selected for the experiments?

The commercial sources of these proteins are noted in the table below and this information has now been incorporated in the ‘Resources’ table in the methods section of the manuscript.

Protein	Commercial source	Catalogue
Recombinant Human SPARC Protein	R&D Systems	941-SP-050
Recombinant Human IGFBP-5 Protein	R&D Systems	875-B5-025
Recombinant Human GAS6 Protein	R&D Systems	885-GSB-050
Human Plasma Fibronectin Purified Protein	Sigma/Merck	FC010-1MG

The product datasheet for SPARC states there is a predicted molecular weight of 34kDa and an actual mass of 40-50 on SDS-PAGE. This suggests glycosylation, but does not confirm this. Using the N-Glycosylation sites prediction tool NetNGlyc-1.0 (<http://www.cbs.dtu.dk/services/NetNGlyc/>), there are 2 potential glycosylation sites (highlighted in blue) but only one predicted to be glycosylated (Asp at position 116 in red). This would explain the higher mass (40 – 50 kDa) detected on SDS PAGE gel.

Name: NP_003109.1 Length: 303
MRAWIFFLLCLAGRALAAPQQEALPDETEVVEETVAEVTEVSVGANPVQVEVGEFDDGAE
ETEEVVAENPCQNHCKHG 80
KVCELDE**NNT**PMCVCQDPTSCPAPIGEFEKVC**SND****NKT**FDSSCHFFATKCTLEGTKKGHL
HLDYIGPCKYIPPCLDSEL 160
TEFPLMRDWLKNVLVTLYERDEDNLLTEKQKLRVKKIHENEKRLEAGDHPVELLARDFE
KNYNMYIFPVHWQFGQLDQ 240
HPIDGYLSHTELAPLRAPLIPMEHCTTRFFETCDLDNDKYIALDEWAGCFGIKQKDIDKDLVI
..... 80
.....N..... 160

..... 240
 320

(Threshold=0.5)

SeqName	Position	Potential	Jury	N-Glyc agreement result
NP_003109.1	88	NNTP	0.1757	(9/9) ---
NP_003109.1	116	NKTF	0.6864	(9/9) ++

The concentrations of SPARC tested were 1 μ g, 10 μ g and 20 μ g. This range was selected according to published literature cited on line 397, page 10. A concentration of 20 μ g was selected as this concentration significantly influenced cell migration when we analysed the data. The following figure depicts the other concentrations tested and has been added in Extended data Fig. 7d.

Reviewer #2 (Remarks to the Author):

The manuscript by Hewitt et al. provides a variety of state of the art techniques used to evaluate differences in cell behavior in normal versus IPF conditions. Specifically, activities associated with KRT5+ basal cells was addressed in terms of responses to changes in extracellular matrix. As fibrosis is a hallmark of IPF, this is a relevant and important aspect to address. The images and the quality of the movies were exceptional and add nicely to the paper. The technological advances used to address ECM composition, organization, and cell response to ECM are laudable. While the results are interesting and contribute to the field, there are a number of critically important aspects to address prior to publication.

1) Fig. 1: Was any attempt made to localize SPARC in tissue sections from IPF lungs? One would perhaps anticipate that higher levels of SPARC would be found in regions where KRT5+ cells associate with fibrosis, i.e. honeycomb and peribronchial areas. There are a number of anti - SPARC antibodies available that work well in human tissue sections (e.g., Sweetwyne et al, JHC 2016).

Thank you for this comment. We attempted to stain for SPARC in lung tissue sections from IPF patients using a commercially available antibody; Recombinant Anti-SPARC antibody [SP205] (ab225716) from Abcam. The lung tissue sections stained were from FFPE blocks therefore it was necessary to perform antigen retrieval following dewaxing and rehydration. Despite trialling different antigen retrieval buffers and heat mediated retrieval methods, there was background nonspecific staining and when compared to isotype and secondary- only controls there was no difference in staining. As the images obtained were substandard, we decided not to include them in the manuscript. Although the reviewer highlights a paper by Sweetwyne et al, JHC 2016 where anti- SPARC antibodies were used in skin and testes using a peroxidase immunohistochemistry protocol, they did not test the antibodies with immunofluorescence. In our hands, immunofluorescence staining of lung did not clearly delineate positive versus negative staining, therefore we did not include in the manuscript. Kuhn et al. 1995 (ref. 44) showed SPARC localised intracellularly to fibroblasts within fibrotic foci however the antibody used for this study is no longer available. Others have shown SPARC in the secretome of IPF fibroblasts (Conforti et al. 2020, ref. 65).

Exploring the Kaminiski/ Rosas single-cell RNA-seq dataset on www.ipfcellatlas.com we see increased expression of SPARC in aberrant basaloid cells, fibroblasts and myofibroblasts derived from lung tissue (see Figure below).

Figure depicting expression of SPARC by cell type from single-cell data at www.ipfcellatlas.com.

2) Fig 2: Was any attempt made to localize fibronectin in IPF vs. normal lungs as fibronectin appears to provide the best substrate for KRT5+ cell migration in CDM (as Fig. 5)?

We localised fibronectin in normal and IPF lung tissue sections using IMC and have now included this in Figure 2 with reference to it in the results section, ‘Distinct regional ECM organisation in the fibrotic lung influences KRT5+ basal cell distribution’ (lines 175 – 177, page 5, marked copy).

3) Fig. 4: The most critical weakness in these studies is the lack of ascorbate in the generation of CDM. The methods state that ascorbate was not added to cultures because ascorbate would promote cross-linking of collagen and that this would artificially alter the properties of the matrix in a way that would influence cell migration or MS-proteomic results. First, because all fibrillar collagen is cross-linked in vivo, as this is a requirement for incorporation into an insoluble ECM, this reviewer would argue that a cross-linked ECM is a more valid construction to test cell response to ECM. But, more importantly, ascorbic acid is not primarily a mediator of collagen cross-linking. The mechanism of ascorbic acid is to serve as a co-factor for prolyl hydroxylase. Hydroxylation of prolines is required for efficient post-translational modification of not only fibrillar collagens, but basement membrane collagen IV as well. In the absence of sufficient ascorbic acid, collagens can be retained in the ER and Golgi and subject to increased intracellular degradation (this is true for all

collagen producing cells and was specifically addressed in human lung fibroblasts by Berg et al. Arch. Biochem. Biophys. 1983). Less collagen will be secreted and less properly modified collagens will be available for proper incorporation into the ECM. This will make for a very different ECM composition than what is produced in vivo. The lack of ascorbic acid provided to cells generating CDM is considered to be a significant flaw and hampers the interpretation of the experimental results of these studies.

We acknowledge the important role played by ascorbic acid in secretion and assembly of human collagens in CDMs and we have included a statement on this (see below). However, this study was not aimed at collagens specifically but rather the production of multiple matrisome proteins. Furthermore, our primary focus was to establish differences in the IPF and control matrices and assess the way these influenced cell migration, irrespective of modifications introduced by ascorbic acid supplementation. Therefore CDMs from both cell types were generated under the same conditions and we observed the reported differences. Whilst the secretion and assembly of collagens may perhaps have been inadequate, we do not consider that this has influenced results we observed.

“Ascorbic acid plays an important role in secretion and assembly of human collagens⁸¹. This study was not aimed at collagens specifically but rather the production of multiple matrisome proteins and the way these influenced cell migration, irrespective of modifications introduced by ascorbic acid supplementation. CDMs from control and IPF fibroblasts were generated under the same conditions.”
(Lines 1152 – 1156, page 29, marked copy)

4) Fig. 5, it was difficult to reconcile the differences in panel e, where no significant differences were found in cell displacement, versus panel h. Please provide a better explanation as to how mean square displacement differs from cell displacement and why this would generate divergent results.

Displacement measures the total distance travelled by a cell by drawing a straight line between its origin and location at the end of the 12 hour measurement period. Therefore, a cell may migrate a considerable distance but if it returns close to its point of origin within the 12 hour measurement period, its total displacement measured by line length will be low. Mean squared displacement (MSD) has its origin in particle tracking and is calculated as a sum of the squared distances from the beginning of the track (Wang et al. PloS One (2014)). MSD therefore increases as the time series progresses. From the gradient of the line it is possible to assess whether a particle (or cell) diffuses freely (straight line), is constrained (line plateaus) or is actively transported / trafficked (line increases gradient). As a result of the way MSD is calculated, differences between the two groups become easier to visualise (Fig. 5e).

We have now re-analysed the data, reorganised Figure 5 and included new description of the results to improve clarity for the reader on this point:

“Visualisation of individual cell track trajectories according to CDM type (Fig. 5c) and tracks plotted from centroid (Fig. 5d), suggested reduced KRT5⁺ BCs migration on IPF CDMs compared to control CDMs. In accordance with this, cell MSD over time was reduced on IPF CDMs compared to control CDMs (Fig. 5e). From the gradient of the line it is possible to assess whether a particle (or cell) diffuses freely (straight line), is constrained (line plateaus) or is actively transported / trafficked (line increases gradient). As such it appears that in both groups, cells migrate in an unrestricted manner and therefore comparing displacement at the end of the experiment is an appropriate measure of migration. When KRT5⁺ BCs were placed onto CDM, more than half of the cells were sessile moving less than two average cell lengths (<30µm),

and the remaining cells were either motile (moving up to 90 μ m) or were highly motile (moving >90 μ m) (Fig. 5f). The relative proportion of motile versus highly motile cells differed significantly according to CDM type (Fig. 5f). KRT5⁺ BCs on control CDMs were more likely to be highly motile (Fig. 5f) and so, given this difference in proportion, sessile cells were excluded from further analysis. KRT5⁺ BCs showed a trend to greater displacement (Fig. 5g), track speed (Fig. 5h) and straightness ratio (Fig. 5i) on control CDMs.” (Lines 291 – 313, pages 8 -9 marked copy).

5) Fig. 6: panel g – SPARC is well-characterized as a counter-adhesive protein (e.g. Sage et al, J. Cell Biol. 1989). It is not accurate to place SPARC in the Venn diagram between cell migration and ECM organization with exclusion from cell adhesion.

We acknowledge the role of SPARC as a counter-adhesive protein, however Fig. 6g was created using the results of DAVID GO enrichment analysis using biological processes and SPARC did not feature in the cell adhesion pathway (GO:0007155). To clarify this we have added a title for this figure panel.

6) Fig. 7: what is the rationalization for using only KRT5⁺ cells from normal lung in cell migration assays in Fig. 7? Wouldn't IPF KRT5⁺ cells be the more relevant cell type to use? Or a comparison between the normal and IPF perhaps showing equivalent (or not equivalent) effects in normal versus IPF cells?

We used healthy KRT5⁺ cells for the cell migration experiments detailed in Fig.7 because we had previously demonstrated no significant migration differences between IPF and healthy KRT5⁺ BCs (Fig. 3 and Fig. 4). The focus of this experiment was therefore to assess the influence of matrix properties on basal cell phenotype, rather than intrinsic disease state of KRT5 basal cells.

“We have shown that KRT5⁺ BC migration is influenced by differences in the properties of the ECM microenvironment and not cell intrinsic disease state. We next sought to functionally validate the influence of mass spectrometry-identified ECM proteins on healthy KRT5⁺ BC migration.” (Lines 391 – 394, page 10 marked copy)

7) SPARC is a collagen binding protein. That addition of SPARC to cells plated and migrating on collagen would impede collagen-dependent migration is not particularly surprising. That the Discussion states that, “Our data are the first to show a function for SPARC in human tissue” is over-stated. First, a function of SPARC in human tissue was shown for by Mendoza-Londono et al (Am J Human Gen. 2015) where missense mutations in SPARC were shown to be causative in osteogenesis imperfecta. Second, showing that SPARC inhibits collagen-dependent migration in an in vitro assay is not conclusive of a function of SPARC in human tissue. Perhaps, recommend modifying the statement to reflect that the data strongly support a functional activity for SPARC in lung/soft, connective tissue, for the first time.

Thank you for this suggestion, we have modified the text accordingly:

“Our data strongly support a function for SPARC in human lung tissue and fibrosing lung disease.” (Lines 483 – 484, page 13 marked copy)

We have also changed the text in the results section along similar lines:

“Together these results indicate that SPARC is a unique regulator of KRT5⁺ BC migration. An abundance of SPARC in IPF CDMs restricts KRT5⁺ BC movement and

its deposition by IPF HLFs represents a mechanism that could explain the retention of KRT5+ BCs in the fibrotic niche.” (Lines 410 – 413, page 11, marked copy)

Minor points:

- a) The source of recombinant SPARC is not given in the list of reagents.

We have now added details of this to the resources table on page 41.

- a) The Biorender generated diagrammatic panels do not lend a great deal to the figures.

Given the complex nature of the experiments we feel the Biorender panels do provide a helpful overview for the reader so we have decided to keep these in.

- b) Fig. 7, panel G – the quality of this summary figure could be improved. For example, at one point in the manuscript, the authors mention cell:cell close proximity and perhaps contact with KRT5+ cells and fibroblasts. As should be indicated, the ECM is a complex composition of proteins, why would the summary figure depict a grossly simple structure with the word SPARC interspersed in the gray lines when the entire paper is devoted to a better understanding of the complexities of the ECM?

We agree with this comment and have decided to remove this figure from the manuscript.

Reviewer #3 (Remarks to the Author):

In their study “Lung extracellular matrix modulates KRT5+ basal cell activity in pulmonary fibrosis” by Hewitt et al., the authors investigate the migratory behaviour of KRT5+ (basal) cells (BC) in healthy lung tissue and that of patients with idiopathic pulmonary fibrosis (IPF). To this end, they use a variety of methods to assess the structure and composition of the extracellular matrix including 2D and 3D migration assays, second harmonic generation 2-photon microscopy, imaging mass cytometry and mass cytometry. In their study, they find differences in collagen deposition and structure between the different disease conditions and observe different migratory behaviours between KRT5+ BC and KRT5+ cells found in IPF. They then also compared the gene expression profiles of healthy and IPF KRT5+ cells when grown on either collagen I coated 2D assays or 3D cell-derived matrix and observed major differences. Notably, collagens were upregulated in all KRT5+ cells cultured on IPF derived matrix. Assessment of matrix derived from healthy and IPF tissues then showed the upregulation of 14 matrix proteins associated with matrix remodelling. The effect of five of these 14 upregulated matrix proteins on KRT5+ cell migration was then assessed in 2D migration assays. This revealed the inhibitory effect of SPARC1 on the migration of KRT5+ basal cells. The authors show that different ECM compositions appear to have an effect on the migratory behaviour of KRT5+ cells, however there are several concerns regarding this study, especially regarding the cogency of experiments from which the conclusion is being drawn that SPARC is the key player in KRT5+ cell migration in IPF.

Major concerns:

1. Generally, formal statistical comparisons are missing, often between IPF and healthy control tissue. This especially concerns figures: 1c,3i In lines 126-127 it is stated that “higher collagen density was evident “. However, while other significant p-values are marked on the figure, this comparison was either not included or was not significant.

Thank you for this comment. In order to address this valid point, we engaged specialist statistical expertise via Dr Ed Roberts (Beatson Institute) who re-evaluated our data and ensured statistically robust methodology were used.

For Figure 1c, honeycomb cysts (HC) and fibrotic regions were present in lung tissue from IPF patients but absent in control tissue therefore limiting direct comparison of these regions. Higher collagen density was consistently evident throughout IPF lung tissue samples. We agree this sentence is unclear so have adjusted the wording as follows:

“A trend towards higher collagen density was evident in the IPF peribronchial and alveolar regions compared to control which did not meet statistical significance after correction for multiple comparisons (Fig. 1f). Fibrotic and HC regions were collagen-dense and present exclusively in IPF lung tissue.”
(Lines 128 – 132, page 4, marked copy.)

For Fig. 3i we did not find any significant statistical difference when plotting IPF vs. control for each ECM ligand. These figures are included below for reference but have not been included in the manuscript as they would detract from the key message of this figure which is that the ECM influences BC migratory dynamics. At the discretion of the Editor we can include these figures as extended data but this would be repetition of data.

To improve the clarity and logical flow of statistical comparisons in other figures related to cell migration experiments, we have re-ordered the following Figs. 3f – i, Figs. 4c-g and Figs. 7a-f. This allows the reader to appreciate the individual track trajectories and mean squared displacement over time before viewing formal statistical comparisons for total displacement, track speed and straightness. The text corresponding to these figures in the results sections has been adjusted to reflect these changes and is detailed below:

“BC mean squared displacement (MSD) over time (Fig. 3f) and total cell displacement (Fig. 3g) was greater on collagen I and III compared to collagen IV and versican. Mean track speed (Fig. 3h) and straightness (or directionality) ratio (Fig. 3i) were also significantly increased on collagen I and III compared to collagen IV and versican.” (Lines 218 – 223, pages 6 – 7 marked copy).

“Cell tracks plotted from centroid showed overlap between healthy and IPF KRT5+ BCs (Fig. 4c). There were no significant differences in MSD over time (Fig. 4d), total displacement (Fig. 4e), track speed (Fig. 4f) or track straightness (Fig. 4g) between healthy and IPF KRT5+ BCs on IPF CDMs.” (Lines 251 – 264, pages 7-8 marked copy)

“When individual cell tracks were visualised (Fig. 7a) and plotted from centroid (Fig. 7b), the addition of SPARC to collagen I appeared to restrict KRT5+ BC migration. When quantified, SPARC caused a reduction in cell MSD over time (Fig. 7c) and total displacement (Fig. 7d). KRT5+ BC track speed (Fig. 7e) and straightness ratio (Fig. 7f) were lower with SPARC.” (Lines 405 – 409, page 11 marked copy)

2. In figure 1d, it is not clear from the text which different conditions or tissue sites are being referred to. In lines 127-129 it is stated that “IPF HC regions clustered with the peribronchial regions of IPF and controls due to increased collagen density (Fig. 1e) and coherency of fibre orientation (Fig. 1f) within these areas.”. IPF HC regions are not labelled on the corresponding PCA plot in Figure 1d.

IPF HC regions were labelled as ‘cysts’ in the PCA plot. These regions are not present in the healthy lung. We have updated the Figure 1d with new group-region labels: ‘CTRL-ALV’, ‘CTRL-PB’, ‘CTRL-PVS’, ‘IPF-ALV’, ‘IPF-FIBRO’, ‘IPF-HC’, ‘IPF-PB’ and ‘IPF-PVS’. ‘IPF HC’ to make this clearer.

3. In figure 1h, Pearson correlation is being used to correlate the number of KRT5+ cells /mm² with collagen density. ...

Thank you for highlighting this. The data is not from a normal distribution so a nonparametric Spearman correlation was used instead. The significance of the correlation remains. This has been updated in Figs. 1i and 1j.

4. Imaging mass cytometry was used to acquire multiplexed images of one healthy control sample and one IPF sample. KRT5+ is being used to identify KRT5+ (basal) cells. Yet, a nuclear marker is missing to properly identify cells within the measure area. Given the highly multiplexed nature of the method, it would have been desirable to include more markers for both the ECM as well as other cell types in the tissue, including fibroblasts, which were identified to be vital for KRT5+ cell migration by the authors. Despite their efforts to visualise SMA+ fibroblasts and KRT5+ cells using high-resolution microscopy, this analysis could be done more efficiently by studying the neighbouring cells of KRT5+ cells between healthy and diseased tissues, by performing a neighbourhood analysis for example. Otherwise, the statement in lines 165-166 “Signalling between these neighbouring cells may drive pathology in the IPF lung.” remains highly speculative. Many more ECM proteins are identified as important throughout this study, including FN1 for example. Thus, it would be interesting to see these markers included for ICM imaging. It would also be interesting to analyse whether the described effect of ECM protein distribution patterns is valid across the entire cohort of patients.

We have included a nuclear marker (Cell-ID Intercalator-Ir; Standard Biotoools; Product ID: 201192A) in Extended Data Fig. 3b. We have now included fibronectin in Fig. 2a-d). We also added Vimentin – a mesenchymal cell marker found in fibroblasts that has been demonstrated to be upregulated at the periphery of fibroblastic foci (Surolia et al. 2019) (Extended data 4c and 4d). We have added text to reflect this in the results section:

“Vimentin is a major cytoskeletal component expressed by mesenchymal cells such as fibroblasts²⁷, and immunostaining of the IPF lung revealed it was upregulated in the periphery of fibroblastic foci²⁸. We found co-localisation of vimentin+ cells and KRT5+ cells in fibrotic alveolar regions (Extended data Fig. 4c) and HC regions (Extended data Fig. 4d).” (Lines 187 – 191, page 6 marked copy).

Ideally, multiple fibroblast markers would have been used across an increased cohort of cases. In the present study, the number of historical lung tissue samples suitable for IMC and the cost of the technique prohibited analysis in larger number of samples.

5. Despite it being mentioned several times later on in the manuscript, it is unclear why FN1 was not included in the different conditions for the migration assay.

We did study KRT5+ BC migration on fibronectin coated plates however it caused high levels of cell proliferation making automated cell tracking impossible to perform accurately. As a result, we were not able to include in the final analysis. We have included a sentence stating this explicitly in the text:

“Fibronectin was not used for the cell migration assay as this promoted significant BC proliferation prohibiting accurate cell tracking analysis.” (Lines 211 – 212, page 6 marked copy)

6. It is unclear to the reader whether the migration assays were done repeatedly and how different many patient samples were involved.

KRT5+ BCs isolated from 5 patients with IPF and 5 healthy controls were analysed. The mean number of cells tracked per healthy subject was 150 and per IPF subject was 213. For the cell migration experiments, 2 or more technical well replicates were performed per donor, and results validated by performing biological replicates.

7. For the 2D migration assays, all 3 collagens and Versican were tested individually. Seeing how the ECM is a mixture of a multitude of matrix proteins, it would be interesting to test the individual behaviours against a mixture of all four components, perhaps even with varying concentrations. (Figure 3e). In figures 3f-h, a formal comparison between IPF and healthy samples would be desirable.

Our aim for the 2D migration assays was to establish the individual contribution of each ECM ligand on KRT5+ BC migration. Whilst it could be interesting to modify the concentrations of each ECM ligand in a mixture, this would be artificial, and we question the in vivo relevance and translational value of this approach. Rather, we chose a 3D CDM model as we felt this would be a more relevant model to assess the impact of a mixture of ECM proteins on migratory dynamics. For Fig. 3 a direct comparison between IPF and healthy

samples for each ECM ligand was performed and there were no significant differences. This data is included above in response to comment (1). We have plotted the data separately for IPF and healthy controls for clarity.

8. Since the KRT5+ cells show different migration patterns in 2D and 3D, it would be interesting to know whether the KRT5+ cells also migrated through the matrix in the 3D assay.

Our live cell microscopy demonstrated in supplementary videos 2,3 and 4 show KRT5+ cells migrating through the matrix in the 3D assay.

9. Seeing how KRT5+ cells also contribute to the ECM production (Figure 4k), it is unclear to the reader, if the described effects of changes in the ECM composition are related to the KRT5+ cells themselves or really the surrounding fibroblasts.

Our proteomic data (Fig.6) shows distinct differences in ECM composition produced by cultured fibroblasts from IPF and control subjects. There were no KRT5+ BCs in this culture system therefore they would not be directly contributing to any changes in ECM production. We and others (Hackett et al. 2011) have demonstrated overexpression of various ECM components by KRT5+ BC, but we did not assess the ECM secretome of KRT5+ BCs within the present study.

10. It is unclear why 2D assays were chosen in figure 7 to assess the impact of SPARC1 on KRT5+ cell migration and why this experiment was not repeated with IPF derived KRT5+ cells.

2D assays were chosen to avoid confounding from other properties of the 3D matrix that can influence migration. We used healthy derived KRT5+ basal cells because our data did not show major differences between KRT5+ BCs driven by intrinsic properties / disease state and our overriding hypothesis – supported by our findings - is that the matrix microenvironment drives abnormal phenotype in this cell type. We have now included a sentence to clarify this point in the text:

**“We have shown that KRT5+ BC migration is influenced by differences in the properties of the ECM microenvironment and not cell intrinsic disease state.”
(Lines 391 – 392, page 10 marked copy)**

11. It is hypothesised that SPARC is being produced by human lung fibroblasts. However, functional assessment is missing to support this hypothesis. Since the authors use fibroblast derived matrix in their experiments, this **could** be further analysed using SPARC knock-out models.

MS-proteomics demonstrated that SPARC was increased in CDM synthesised by IPF fibroblasts compared to that produced by control lung fibroblasts. Moreover, our in vitro analysis revealed that there was an effect on cell migration when we performed a migration assay. SPARC reduced migration of KRT5+ basal cells. We wanted to focus our studies on human primary cells, given that there are well- described limitations of the currently available mouse models of pulmonary fibrosis (**Jenkins et al. Am J Respir Cell Mol Biol 2017**).

In the discussion we reference previous studies (Chang et al. 2010, Conforti et al. 2020) which demonstrate SPARC is increased in conditioned media from IPF HLFs. In vivo models of bleomycin-induced pulmonary fibrosis demonstrate reduction in

collagen production in SPARC-null mice. We have added a sentence to reflect this in the discussion:

“In vivo models of bleomycin-induced pulmonary fibrosis demonstrate a reduction in lung collagen content in SPARC-null mice⁶⁸ and following SPARC siRNA treatment⁶⁹.” (Lines 481 – 483, page 13 marked copy.)

Minor concerns:

1. Instead of only showing healthy control lung samples in the supplementary figure, it would be beneficial for direct comparison to show both healthy and diseased tissue in figure 1. It would further also be helpful to the reader to label the healthy and diseased tissue images as such in the 1§.

We have now added this comparison in Fig. 1b and 1c.

2. Figure legends are often only displayed once in a figure with multiple panels but should be added to each figure panel for easier and faster readability.

We have updated this to ensure legends are displayed for each panel.

3. The genes in figures 4j and 6e are not readable.

We agree the gene names in Figs. 4j and 6e are very small but we were constrained by space. These heatmaps are both high- resolution and we found the genes to be visible when using the zoom function on PDF. If the Editors permit, we could increase size/ resolution in a separate supplementary figure.

4. It is also unclear what each cluster is formed of in figure 6e. For better readability, it would be easier to include a .csv file with all differentially expressed genes per cluster and to highlight only the most important ones per cluster. There are also considerable differences between the clusters which are not explained. In figures 6e-f, a legend is missing explaining the different colours. The heatmap counts bar is also missing a label.

We have now improved the readability of this figure by addressing these points. We have added a .csv file with the raw data used to generate this heatmap. A legend has now been added for Figs. 6e-f to explain the different colours. The heatmap counts bar is described in the figure legend.

“Heatmap with hierarchical clustering of normalised abundance of each protein identified. Rows and columns using correlation distance and average linkage. Rows are centred and unit variance scaling is applied.” (Lines 844 – 846, page 21 marked copy)

5. Figure 6d is barely readable.

To improve readability, we have increased the figure and font size.

6. Figure 7e is missing a legend and axis. All other migration assay plots are also missing axis descriptions.

In 7e, all tracks are plotted from centroid. This form of plot has been used for migration assays in other published work (Puttur et al., Sci. Immunol. (2019)). Whilst there is no specific axis for this type of plot, we have now demarcated

the perimeter. The scale used for each of these plots is detailed in the figure legend.

7. Capitalisation is inconsistent throughout all figures (e.g., log2 (Figure 6b)– Log2 (Figure 6f))

We have now addressed this to ensure capitalisation consistency in all figures.

REVIEWERS' COMMENTS

Reviewer #1 (Remarks to the Author):

The concerns raised were addressed appropriately.

Reviewer #3 (Remarks to the Author):

The authors have improved the manuscript, have addressed the main points of the reviewer and it has become much clearer, especially regarding all statistical analyses and the overall structure.

Perhaps our comment on nuclear markers regarding the images in figures 2 and extended data figure 4 was not very clear in the first round of review. Please add the nuclear DNA staining to these figures as done with DAPI or Histone H3 in all other images in this manuscript (including IMC images in extended data figure 4c-d).